

# Estimating water flux and evaporation losses using stable isotopes of soil water from irrigated agricultural crops in tropical humid regions

Amani Mahindawansha[1,2], Christoph Külls[4], Philipp Kraft[1], and Lutz Breuer[1,3]

[1]Institute for Landscape Ecology and Resources Management (ILR), Research Centre for BioSystems, Land Use and Nutrition (iFZ), Justus Liebig University Giessen, Giessen 35392, Germany
[2]International Rice Research Institute (IRRI), Los Baños 4030, Philippines
[3]Centre for International Development and Environmental Research (ZEU), Justus Liebig University Giessen, Senckenbergstrasse 3, Giessen 35390, Germany
[4]Faculty of Civil Engineering, Technische Hochschule Lübeck, Mönkhofer Weg 239, Lübeck 23562

*Correspondence to*: Amani Mahindawansha (Amani.Mahindawansha@umwelt.uni-giessen.de)

**Abstract.** Reliable information on water flow dynamics and water losses via irrigation on irrigated agricultural fields is important to advance water management strategies. We investigated the effect of season (wet season, dry season), irrigation management (flooded, non-flooded), and crop diversification (wet rice, dry rice, and maize) on soil water dynamics and water losses via evaporation during plant growth. Soil water was extracted and analyzed for the stable isotopes of water ($\delta^2$H and $\delta^{18}$O), and the fraction of evaporation loss was determined using the Craig–Gorden equation. For all crops, shallow soil compartments (0 to 0.2 m) were more isotopically enriched than deep soils (below 0.2 m). The soil in maize fields showed stronger evaporation enrichment than rice, which increased as the crops progressed through the growth; however, it decreased in both rice varieties during both seasons. Greater water loss was encountered during the wet season even though evaporation signals were stronger during the dry season. The enrichment of surface water was reflected in shallow soils of wet rice, and it decreased towards the end of growth during the wet and dry seasons. Isotope data indicated that soil water flow mechanisms varied depending on field conditions. In flooded conditions, surface soil was consistently affected by piston type matrix flow. During non-flooded conditions, matrix flow via diffusion dominated compared to upwards evaporative flux. Occasionally, preferential flows occurred through cracks, especially in maize fields. In wet rice fields, soil water was largely influenced by short-term variability of precipitation events during the wet season and subsequent formation of hydrogen compounds as a result of continued wetness and anaerobic physiochemical conditions that depleted $\delta^2$H with respect to $\delta^{18}$O.

## 1 Introduction

Soil water studies are essential for a better understanding of the role soils play in the hydrological cycle, in order to estimate the water budget and water availability for plants, groundwater recharge, other organisms as well as solute transport (Sprenger et al., 2015; Vereecken et al., 2016). Stable isotopes of water ($\delta^2$H and $\delta^{18}$O) have become a powerful tool for such studies as





ideal natural tracers (Kendall and Caldwell, 1999). Stable isotopes of water have been used to identify and understand environmental processes, including soil water movement and solute transport (Groh et al., 2018), flow pathways and mixing (Mueller et al., 2014; Stumpp and Maloszewski, 2010; Windhorst et al., 2014), groundwater recharge (Beyer et al., 2015; Houben et al., 2014), hydraulic redistribution (Priyadarshini et al., 2016), plant water uptake (Mahindawansha et al., 2018b),

soil water exchange in the deep vadose zone (Gehrels et al., 1998), extreme events such as droughts (Chiogna et al., 2018), and isotopic sub-daily patterns in groundwater and ponding water (Mahindawansha et al., 2018a). In recent years, the interest has shifted towards understanding the evaporation dynamics in soil water (Braud et al., 2009; Kool et al., 2014; Rothfuss et al., 2015), because the composition and distribution of stable isotopes of water in a soil profile provide insight to evaporation fractionation and water flux processes (Wenninger et al., 2010). However, the quantification of soil evaporation processes is

also empirically more demanding (Bittelli et al., 2008), and therefore, the quantification process has not been studied as much (Evaristo et al., 2015; Sprenger et al., 2017).

The isotopic composition of soil water is not only affected directly by evaporation, mixing of new and old water (Gazis and Feng, 2004), and altering input signals (Barnes and Turner, 1998), but it also indirectly by other processes such as transpiration (Barnes and Allison, 1988), water transport (Kutilek and Nielsen, 1994; Melayah et al., 1996), and hydrodynamic dispersion

(Wang et al., 2017). The isotopic enrichment of shallow soil water is generally driven by evaporation during drier periods (Gangi et al., 2015; Liu et al., 2015) and affected by equilibrium and kinetic fractionation (Benettin et al., 2018). Many experiments on the effects of evaporation on soil water using isotope methods are restricted to laboratory-scale or short-term field studies or to one particular location (Gaj et al., 2016; Oerter and Bowen, 2017; Rothfuss et al., 2015; Sprenger et al., 2017; Twining et al., 2006; Volkmann et al., 2016).

Studying rice-based crops is important, because rice (Oryza sativa L.) is the dominant staple food for nearly half of the world's population. More than 80 % of global rice production area is located in Asia (Kudo et al., 2014). It is one of the highest water-consuming grain crops (Janssen and Lennartz, 2007; Mekonnen and Hoekstra, 2011), consuming approximately 30 % of all freshwater resources worldwide (Maclean, 2002). Because rice is extremely sensitive to water deficit (Bouman and Tuong, 2001), 80 % of rice production is cultivated under conventional flooded conditions in Asia (Towprayoon et al., 2005) also

called wet rice, anaerobic rice, or lowland rice. Water scarcity is a serious environmental problem, especially concerning irrigation in agricultural lands (Navarro-Ortega et al., 2015; Pfister et al., 2011). Therefore, water saving strategies need to be developed to ensure safe rice production for future generations (Belder et al., 2004). By introducing non-flooded crops during the dry season (e.g., rotating maize/dry rice with wet rice) is an interesting alternative and has been increasingly applied in food and fodder production in Southeast Asia (FAO, 2016; Timsina et al., 2010). To establish an efficient water-saving

management system based on crop rotation and seasonal changes, and to adapt to the effect of climatic changes, a detailed and functional understanding of hydrological processes and water fluxes in irrigated agricultural systems is necessary (Daly et al., 2004; Heinz et al., 2013; Zwart and Bastiaanssen, 2004).

Understanding water flow dynamics and estimations of evaporation fluxes from irrigated soils in the subsurface is a general fundamental challenge and is poorly understood in hydrological and ecohydrological studies in rice-based cropping systems.



Moreover, studies on the effects of evaporation on the dynamics of stable isotopes of soil water, its temporal (i.e., seasonal) variability, as well as the impact of various crop rotations are still missing. None of the studies conducted so far have quantified the fraction of soil water evaporation in irrigated agricultural fields while also taking into account the effect of crop species and various growing stages.

Our objectives during this study are: (I) to investigate natural soil water isotopic profiles as a function of soil depth depending on season (wet and dry), type and growth of vegetation (wet rice, dry rice, and maize), and differences in irrigation patterns; (II) to understand flow mechanisms and redistribution patterns of soil water in the soil matrix; and (III) to quantify the fraction of soil evaporation losses at different soil depths based on information about both $\delta^2$H and $\delta^{18}$O.

## 2 Material and Methods

### 2.1 Site description and experimental design

The field trail was established at the experimental station of the International Rice Research Institute (IRRI), in Los Baños, Laguna, Philippines (14° 11' N, 121° 15' E, 21 m a.s.l.) and used during both the wet (WS) and dry (DS) season. The average total precipitation was 1,700±50 mm during the WS (June to November) and 300±25 mm during the DS (December to May). The mean seasonal temperature and relative humidity were 28.5±0.9°C and 83±6 % during WS 2015, respectively, as well as

27.6±1.8°C and 74±11 % during DS 2016, respectively. Climate data were obtained from the climate unit at IRRI. The soil type in the study area is classified as a Hydragric Anthrosol (He et al., 2015) with clay-dominated soil texture (Table 1). The clay fraction mainly consists of vermiculite and smectite as three layer clays and kaolinite as a two-layer clay. Three-layer vermiculite is mainly responsible for the swelling and shrinking of the soil matrix (Tertre et al., 2018).

The experiment was conducted during WS 2015 and DS 2016. The experimental design (Fig. 1) consisted of nine fields with

an average field size of about 540 m$^2$, each split into three plots with different treatments (i.e., straw application (S), no straw as a control plot (C), and straw application with mung bean as an intercrop (M)). To maintain constancy for our experiment, plots with mung bean treatment (M) were excluded from sampling, because that treatment was only applied during the transition period between DS and WS. During the WS, all fields were cropped with wet rice (cultivar NSIC Rc222). During the DS, three fields were each cultivated with wet rice, dry rice (cultivar NSIC Rc192), and maize (Pioneer P3482YR). Wet

rice fields were maintained at water-flooded conditions, except for the first and last two weeks between transplanting and harvest. Dry rice and maize fields were only irrigated when weather conditions suggested a water shortage (i.e., 5–10 times during the growing season for maize fields). The total irrigation amount for wet rice fields was 470±50 mm during the WS, and 1,270±300 mm, 517±50 mm, and 212±50 mm for wet rice, dry rice, and maize during the DS, respectively. Transplanting and harvesting dates for rice were July 21st and October 30th during the WS. During the DS, the transplanting date was January

8th, and harvesting dates were April 10th for wet rice and April 17th for dry rice, and January 6th and May 11th for maize in 2016, respectively (Fig. 2).





## 2.2 Soil and root sampling

Samples were collected during the three main growing stages (GS) described by Counce et al. (2000), i.e., at the vegetative stage (GS1, from germination to panicle initiation), the reproductive stage (GS2, from panicle initiation to flowering), and the ripening stage (GS3, from flowering to maturity). The growing stages were used as a reference time scale along with the plant

growth (Fig. 2). Therefore, the growing stages for rice and maize were assumed to be similar to maintain consistency of sampling conditions. The sampling campaigns were conducted one day during each growing stage at 26, 55, 85 days after transplanting during the WS and 40, 60, 90 days during the DS, respectively. For this experiment, during each sampling campaign (3 in a season) 18 soil cores using a manual soil corer (length=0.6 m, diameter=0.05 m) were taken. This is a collection of 2 cores from each plot (only from S and C plots) from 2 seasons (during the WS: wet rice n=18, during the DS:

six samples each for wet rice, dry rice, and maize, in total n=18). Each core was further divided to 9 depth intervals (9 samples from each core) from the surface to 0.6 m (0, 0.05, 0.1, 0.15, 0.2, 0.2–0.3, 0.3–0.4, 0.4–0.5, 0.5–0.6 m). Altogether 108 soil cores were taken during each of the three growing stages and throughout both WS and DS, which gave a grand total of 972 samples. A plastic ring (diameter=0.5 m) was used to drain the water around the sampler prior to coring in wet rice fields. Samples were stored in sealed aluminium bags (CB400–420BRZ, 80 mm x 110 mm, Weber packaging, Güglingen, Germany)

and immediately placed in an ice-filled Styrofoam box for transfer to the laboratory where they were kept frozen.

Soil water was extracted from soil aliquots (10–15 g of the sample) via cryogenic vacuum extraction (Orlowski et al., 2013) at the Institute for Landscape Ecology and Resources Management (Justus Liebig University Giessen, Germany) for four hours at 200°C under a pressure of 0.3 Pa. The gravimetric soil water content along the soil profiles was determined based on the soil weight loss following cryogenic water extraction. Groundwater and surface ponded water were collected once a week

from each plot at existing sampling stations (Heinz et al., 2013). Rainwater and irrigation water were sampled according to their availability. Root length density (cm cm$^{-3}$) was analyzed using the winRHIZO software (WinRHIZO 1991) in the plant physiology lab at IRRI. For detailed information about the experimental design, sample collection, and root density analysis, see Mahindawansha et al. (2018b).

## 2.3 Isotopic measurements

The oxygen and hydrogen isotopic compositions of the water samples (extracted soil water and liquid samples) were measured via off-axis integrated cavity output spectroscopy (OA–ICOS, DLT–100–Liquid Water Isotope Analyzer, Los Gatos Research Inc., Mountain View, CA, USA) and reported in permil [‰]. The analytical precision for $\delta^{18}O$ and $\delta^2H$ was 0.2 ‰ and 0.6 ‰, respectively.

The global meteoric water line (GMWL) was determined following Rozanski et al. (1993) ($\delta^2H = 8.2\delta^{18}O + 11.3$). The Local

Meteoric Water Line (LMWL) was calculated with $\delta^2H = 7.52\delta^{18}O + 5.86$, using stable isotope compositions of local precipitation collected from 2000 until 2015 (GNIP–IAEA, 2016). Line conditioned excess (lc-excess) was calculated for soil water samples as suggested by Landwehr and Coplen (2006) with lc-excess $= \delta^2H - a\delta^{18}O - b$, where a and b refer to the slope





and intercept of the LMWL, respectively. We used the lc-excess to infer seasonal dynamics of evaporation fractionation (Sprenger et al., 2017).

## 2.4 Calculation fraction of evaporation

The joint effect of equilibrium and kinetic isotopic fractionation during the phase transition from liquid water to vapor can be
estimated using the Craig–Gordon model (Craig and Gordon, 1965). Sprenger et al. (2017) have recently used Equation 1 to estimate evaporation from the topsoil (0–0.1 m). We assume that this model controls the development of soil water isotopic composition in the uppermost soil compartment. This isotopic signal is then carried to deeper compartments via leaching. In deeper compartments, mixing with macropore flow from cracks may occur. The concept of multi-compartment transport indicates the history of the evaporation process as well as the depth and degree of isotope signal changes by the preferential
flow. Equation 1 is based on the Craig–Gordon model and formulations introduced by Gonfiantini (1986) to estimate the fraction of evaporation loss ($F_E$) for an isotope mass balance as follows:

$$F_E = 1 - \left[\frac{(\delta_S - \delta^*)}{(\delta_P - \delta^*)}\right]^m \tag{1}$$

where $\delta_S$ is defined as the isotopic signal of the soil [‰], $\delta_p$ is the original isotopic signal of soil water [‰], $\delta^*$ is the limiting isotopic enrichment factor [‰], and m is the temporal enrichment slope [–]. The original isotopic signal, $\delta_p$, in water during
the WS was estimated as the mean isotopic signal from the most frequent large precipitation events, and as the mean of the irrigation water during the DS. We assumed steady state conditions, because the samples were collected between 10–12 a.m., where steady state conditions can be expected in rice fields (Wei et al., 2015). Variables $\delta^*$ and m were calculated following Equations 2 and 3 (as described in Benettin et al. (2018) and Gibson (2016)):

$$\delta^* = \frac{(RH\delta_A + \varepsilon_k + \varepsilon^+/\alpha^+)}{(RH - 10^{-3}(\varepsilon_k + \varepsilon^+/\alpha^+))} \tag{2}$$

$$m = \frac{(RH - 10^{-3}(\varepsilon_k + \varepsilon^+/\alpha^+))}{(1 - RH + 10^{-3}\varepsilon_k)} \tag{3}$$

where $\delta_A$ is the isotopic composition of atmospheric vapor [‰] (calculated according to Benettin et al. (2018), assuming that the isotopic composition of atmospheric vapor is in equilibrium with precipitation), RH is the relative humidity, $\varepsilon_k$ is the kinetic fractionation factor [‰], $\alpha^+$ [–] and $\varepsilon^+$ [‰] are equilibrium fractionation factors. The temperature-dependent parameter $\alpha^+$ was calculated for $\delta^2H$ and $\delta^{18}O$ separately (Benettin et al., 2018). Furthermore, $\varepsilon_k$ was calculated according to Benettin et al.
(2018), presuming diffusive transport in soil pore spaces (Barnes and Allison, 1983). The equilibrium isotopic separation between liquid and vapor was computed as $\varepsilon^+ = (\alpha^+ - 1)10^3$ [‰] (Gat, 1996; Horita et al., 2008. Benettin et al., 2018). The aerodynamic diffusion parameter, n [–], reaches 1 when the soil is dried to residual moisture levels (Mathieu and Bariac, 1996),





presenting turbulent conditions. Therefore, we anticipated that n is 0.5 for wet rice fields with saturated soils (Good et al., 2014), 0.7 for dry rice, and 0.9 for maize.

## 2.4 Statistical analysis

We tested for significant statistical differences in stable isotopes of water ($\delta^2$H and $\delta^{18}$O) during seasons, growing stages, and treatments between all water sources. Normal distribution was tested by the Shapiro Wilk test and homogeneity of variances by the Fligner Killeen test (Python 2.7.10.0). Because of the non-normal distribution of data, we further carried out a non-parametric rank based test (Kruskal and Wallis, 1952) considering no ties. We rejected the null hypothesis that two profiles were significantly different (p≤0.05) referring to different treatments, seasons, and crops.

The isotopic values of the two treatments (S and C) were combined for each crop for further analysis, because there were no significant differences for stable isotopes of water between the fields with the same crop (p>0.05).

## 3 Results

### 3.1 Soil and water isotopic distribution

Both $\delta^2$H and $\delta^{18}$O values of surface and groundwater were higher at the beginning of each season and decreased towards the end. During both seasons, surface water and groundwater showed a relatively similar range of isotopic compositions at each growing stage with no statistically significant differences (Table 2); however, that is not the case from WS to DS. Rainwater was isotopically similar to groundwater and surface water during the WS, unlike during the DS. Although rainwater and irrigation water were statistically similar during the WS, we found significantly different values during the DS.

Figure 3 displays the $\delta^2$H and $\delta^{18}$O together with water content and lc-excess values in soil water as a function of soil depth during GS1, GS2, and GS3 of wet rice during the WS, along with wet rice, dry rice, and maize during the DS with the standard deviation of the replicates. The range of isotopic composition of rainwater and irrigation water defines the water input to the system at each season (average values are presented in Table 2). The isotopic composition of soil water from crops during the DS were statistically different from the WS crops (wet rice). GS2 and GS3 of maize and wet rice were statistically different during the DS, and maize and dry rice were statistically different except for the GS3 of dry rice. The isotopic signals of the soil profiles to a depth of ~0.2 m were highly variable, becoming more stable further below. Therefore, soil water isotopic values can be divided into two categories: shallow soil water from 0 to 0.2 m, and deep soil water from 0.2 to 0.6 m. The shape of isotopic profiles in the shallow soil water changed depending on the crop and growth stage. In the wet rice soil, the isotopic values increased until the depth of 0.05 m and then decreased again to about 0.2 m (Fig. 3a, b, e, f). Interestingly, in wet rice soils, the depth of the highest isotope enrichment which is just below the soil surface, decreased deeper in the soil during the growing period from GS1 to GS3 in both seasons. In contrast, the shape of the isotopic profiles of dry rice and maize follow a different pattern than wet rice, with higher $\delta^2$H and $\delta^{18}$O values at the soil surface and an exponential decrease down to around 0.2 m soil depth (Fig. 3i, j, m, n). The isotopic composition of shallow soil in dry rice fields decreased from GS1 towards GS3,





where the values were stable in maize fields during all phases of plant growth. However, the isotopic values in deep soil were nearly stable in all the profiles regardless the crop during both seasons.

The wetness of the soil can be identified by water content profiles. Maize was characterized by dry conditions at the surface and at shallow depths compared to both rice varieties. However, the highest water content for wet rice was at the surface soil (17.7±1.2 %), and nearly constant below the depth of 0.2 m (12.0±1.3 %) during both seasons. The water content values in dry rice soils were rather evenly distributed along the soil profile except at the soil surface. Soils were getting dryer as maize plant growth progressed, while such clear patterns could not be observed for any other crops. Below 0.2 m, water content profiles for wet and dry rice illustrated a nearly constant value of about 12 %, while water content gradually decreased in maize fields. The lc-excess is a sign of evaporation, with lower values indicating larger evaporative losses. We found an exponential pattern with lower values in shallow soils, particularly for maize, but also, though less apparent, for dry rice soils. This indicates a higher evaporation signal in shallow soils for these DS crops compared to the WS crop. The most evaporation was found near the surface in maize fields with significantly lower lc-excess values; in addition, lc-excess values further decreased from GS1 to GS3. In contrast, lc-excess patterns at shallow soils for wet rice fields generally increased with growth during both seasons, similar to dry rice fields during the DS (except for GS2). For WS wet rice, we even observed decreasing lc-excess along with the profile at shallow depths. In contrast to other crops, this trend then reverted to a gradual increase in deep soils. The lc-excess values in deep soils increased with the growth of rice but decreased with the growth of maize.

Existing soil water was mixed with the isotopic input signal (i.e., precipitation and irrigation), and therefore had a potentially different isotopic composition than the input water. The $\delta^2$H and $\delta^{18}$O values of soil water and source water plot on a line below the LMWL in the dual isotopic coordinate system ($\delta^2$H, $\delta^{18}$O) due to the evaporation effect (Fig. 4). The slope of the regression line and coefficient of determination ($R^2$) were higher during the DS (avg. slope=5.1, $R^2$=0.92) than during the WS (avg. slope=3.5, $R^2$=0.54). Soil water $\delta^2$H and $\delta$18O compositions were higher (enriched) in shallow soils and more deviated from LMWL than soil water in deep soils.

Original water inputs to the system had changed depending on the season, especially during the WS. As a result of frequent precipitation events introducing strong variations in isotopic composition ($\delta^2$H from –55.20 to –10.89 ‰ and $\delta^{18}$O from –7.91 to –2.54 ‰), the isotopic signal of the input water varies significantly (Fig. 2). We observed lower slopes and more clustered data points in wet rice soil during the WS, indicating lower soil evaporation compared to the DS. During the WS, there are some shallow soil isotopic values plotted close to the LMWL, and some deep soil values deviate more from the LMWL (Fig. 4a–c). This indicates the movement of isotopic signals stemming from the previous DS to deeper compartments of the soil profile. During the DS, slopes of the regression lines were lower for wet rice (slope=5.2, $R^2$=0.88) than for dry rice (slope=6.0, $R^2$=0.94) and maize (slope=5.5, $R^2$=0.91) (Fig. 4d–l). Due to less frequent and short precipitation events during the DS, the original water input to the system was dominated by irrigation water, with nearly constant isotopic composition during the growing period. Small precipitation events were subject to higher evaporative loss and resulted in enriched isotopic composition during this time (Table 2).



## 3.2 Fraction of evaporation estimation

The estimated $F_E$ at each depth was derived by means of an evaporative enrichment of heavier isotopes in soil water (Fig. 5). Soils in dry rice fields showed higher soil $F_E$ at shallow depths (from 0.54±0.1), which decreased both during growth (to 0.27±0.1) and along with depth towards the deep soil (to 0.20±0.1) (Fig. 5g, h, i). Evaporation from soils in maize fields decreased with depth for both isotopes (from 0.31±0.1 to 0.07±0.05) and did not fluctuate significantly during plant growth (Fig.5 j, k, l). The $F_E$ at shallow soils of wet rice ranged from 0.42±0.08 to 0.20±0.08 (similar for both isotopes), and remained nearly stable in deep soil at 0.13±0.1 (Fig. d, e, f). However, the fractionation was higher during the WS, and the $F_E$ for $\delta^2$H and $\delta^{18}$O expressed a significant difference (Fig. 5a, b, c), in clear contrast to data from the DS. During the WS, $F_E$ in shallow soil decreased from 0.72±0.12 (GS1) to 0.47±0.06 (GS3) for $\delta^2$H and from 0.87±0.07 (GS1) to 0.76±0.07 (GS3) for $\delta^{18}$O, while the fractionation was lowest during GS2 for both isotopes. Pore water indicated lower $F_E$ in soils below 0.4 m during the GS1 in dry and wet rice, and this depth decreased to about 0.35 m during GS3. However, there was a clear decrease in the extent of evaporation with growth at rice fields. During the WS, the soil water in wet rice carries a signal of high evaporation losses down to 0.5 m. The estimated $F_E$ from ponding surface water was found to be higher during the WS than during the DS with no significant difference between $\delta^2$H and $\delta^{18}$O. The $F_E$ of ponded water during the WS did not fluctuate with time, and remained close to 0.92±0.07, while during the DS values decreased from GS1 (0.67±0.03) to GS3 (0.24±0.01). Thus, surface water $F_E$ indicates higher evaporation losses during the WS, and the evaporation signal is carried to deeper layers by subsequent percolation.

## 4 Discussion

### 4.1 General mechanisms in soil water movement

The soil of the wet rice fields was mostly saturated by flooding, while the water saturation at the dry rice fields varies greatly with irrigation and precipitation events (Fig. 2). Soil moisture at the infrequently irrigated maize fields was the lowest throughout the cropping season. Depending on the evaporation effect on soil water isotopic composition and water transport, the soil profile can be subdivided into two parts (Barnes and Allison, 1984): (I) shallow soil in which water moves by vapor diffusion and is affected by evaporation, (II) deep soil, in which liquid transport dominates and is barely affected by evaporation. This isotopic separation developed predominantly due to the existence of the dense, least permeable plough pan, which separates the puddled shallow soil and non-puddled subsoil in paddy fields; it is a result of repeated ploughing over many years due to the cultivation (Chen and Liu, 2002). Three general mechanisms can explain water movement phenomena in irrigated fields: (I) matrix flow via diffusion, (II) fast percolation of water through desiccation cracks/deep roots, and (III) continuous slow infiltration from the liquid phase through the clay matrix in flooded fields.

Precipitation and irrigation events partially penetrate down to the unsaturated zone and are then consumed gradually by evapotranspiration (Barnes and Allison, 1988). Therefore, soil water isotopic profiles reflect a balance between water



infiltration (input) and soil evaporation (output) (Hsieh et al., 1998), the latter being responsible for kinetic separation (Barnes and Allison, 1984). Soil water isotopes are affected by an evaporation process in which vapor transport is dominant (Bittelli et al., 2008), especially in dry rice and maize fields. This leads to build-up of heavy water molecules (formed by $^2$H and $^{18}$O) at the water–air interface, which are transported downwards and then mixed with the soil matrix (Horita et al., 2008).

Downward water movement at steady state or slowly changing conditions results in an exponential evaporation profile along the depth during the drying stage that is comparable to those found in dry and maize soils (Fig. 3i, j, m, n) (Zimmermann et al., 1966; Barnes and Allison, 1988; Rothfuss et al., 2015). The downward flow can be via advection, hydrodynamic dispersion (Leibundgut et al., 2009), diffusion (Barnes and Allison, 1983), or preferential flows, which affect the isotopic distribution within the profile in the unsaturated zone (Koeniger et al., 2016). The observed smoothing of isotopic signals in shallow soils

can be explained by water redistribution via shallow roots/transpiration or from preferential flow transferring and mixing the evaporated surface water into deeper soil compartments (Baram et al., 2013).

In maize fields (below 0.2 m), there were deeper (~0.2 m) and narrower (~0.02 m) desiccation cracks than those found in dry rice fields (own observation). However, desiccation cracks in dry rice fields were not as hydraulically active as in maize fields due to differences in irrigation practice (Fig. 2). Therefore, the dominant flow mechanism in maize fields is controlled by

preferential flow through desiccation cracks. During irrigation, water flowing through preferential flow conduits transports and redistributes evaporated water, affecting the capillary gradient between the soil matrix and crack walls. However, water loss from the crack surface is limited by water movement through the soil matrix and higher relative humidity during the night (Kamai et al., 2009; Weisbrod and Dragila, 2006). There was a gradual isotopic depletion towards the deep soils of dry rice and maize fields. This indicates subsurface mixing between enriched soil water and depleted irrigation water that percolated

into the deep vadose zone via preferential flow paths (Baram et al., 2012; Nativ et al., 1995). Baram et al. (2012) have observed that naturally formed desiccation crack systems can create preferential flow paths that reach more than a meter deep. In maize fields at our study site, we observed that the groundwater isotopic compositions are strongly influenced by irrigation water suggesting the existence of fast flow conduits (Mahindawansha et al., 2018a). In addition, He et al. (2017) have observed leaching losses of water and nutrients in a lysimeter experiment. Significant capillary rise is expected in fine textured soils,

and therefore the capillary rise of depleted shallow groundwater can also influence compartments at greater depth (Baram et al., 2013; Clark and Fritz, 1997), even though the groundwater level was below sampling depth (below 0.6 m).

For the constantly flooded condition of wet rice, continuous slow water percolation is observed as expected. The upper soil layer is affected by isotopically enriched liquid phase via a gravity-driven, piston-like matrix flow. The isotopic composition of soil water increased with depth (until the most enriched point) (Fig. 3a, b, e, f). It is assumed that this observation is a result

of the successive displacement of pre-existing mobile soil water by infiltrating water. Still, soil water in fine pores represents quasi-stationary storage exchanging water and isotopes with the mobile phase (Gazis and Feng, 2004). As a result, the ponding water column and the soil water at shallow depth down to the infiltration front, act as a single compartment reflecting evaporation from the ponded water. Isotopic values below this point show a strong depletion until reaching a stable value





below approximately 0.2 m. A similar pattern has been found by Baram et al. (2013) in clay soil under continuous ponded infiltration in Israel.

Systematic isotopic depletion and increasing negativity of lc-excess profiles indicate less evaporation effect from GS1 to GS3 in rice (Fig. 3). In both rice varieties, the isotopic profile showed a clear shift from enriched to depleted values, especially in
shallow soils regardless of the season. We observed a transfer of the most isotopically enriched depth in wet rice down to greater depths in conjunction with plant growth (Fig. 3a, b, e, f). Therefore, we can assume that there is an influence of the crop type and growth stage on evaporation fractionation in the soil water. It was previously shown that the plant cover generally reduces kinetic fractionation processes in soil water (Burger and Seiler, 1992; Dubbert et al., 2013). A different pattern of lc-excess was observed in maize fields (Fig. 3p) compared to rice (Fig. 3d, h, l), in which the evaporation fraction gradually
increased towards the end of the season, resulting in dryness and water deficit as irrigation diminishes (Fig. 2). Finally, kinetic fractionation was diminished by soil dryness resulting from infrequent irrigation.

Comparison of regression lines of soil water samples to the GMWL in dual isotope space ($\delta^{18}$O, $\delta^{2}$H) helps to identify the environmental conditions during soil evaporation with regard to season and crop (Fig. 4). The slope of the $\delta^{18}$O–$\delta^{2}$H relationship decreases as a result of kinetic fractionation (Sprenger et al., 2016). This deviation can then be used to estimate
evaporation losses (Clark and Fritz, 1997). The higher slopes of the dry soils (in maize and dry rice) can be explained by an increase in the effective thickness of the vapor transport layer (Barnes and Allison, 1988) compared to wet soils (as in wet rice). For soils under wet rice, a steeper gradient near the surface was found, similar to observations made by Allisons (1982) for saturated soils. Deep soil water under wet rice exhibits isotope data falling further below the LMWL during the WS from GS2 and GS3 (Fig. 4b, c), indicating higher soil evaporation. In contrast, shallow soils plotted closer to the LMWL indicate
lower evaporation rates. Furthermore, deep soil water shows isotopic similarity to the irrigation water. Following these observations, we can assume that the deep soil isotopic profile results from mixing between irrigation water from the previous DS (memory of the old isotopic signal) that moved downward via matrix flow. Due to low rates of matrix seepage and percolation of 1 to 5 mm d$^{-1}$ in clay soils, (Bouman and Tuong, 2001), deep soil profiles with multiple compartments contain and may reveal a record of antecedent evaporation conditions or preferential flow shortcuts between compartments. However,
all soil profiles present enriched values and significant evaporation during the WS (Fig. 4a–c) as seen by Baram et al. (2013). Lower slopes of evaporation lines in wet soil compared to dry soil point to greater kinetic effects (Cooper et al., 1991). Slopes of evaporation lines <3.5 were observed under diffusion conditions (Allison et al., 1983). Therefore, profiles during the WS indicate that diffusion processes in the subsurface are relevant, especially at GS1 and GS3 (Fig. 4a, c). During GS2, mixing processes between infiltrating water dominate and limit diffusion processes due to continuous intense precipitation events
during that time (Fig. 2). We further observed a higher correlation between plant water and rainwater during this time compared to the other growing stages (Mahindawansha et al., 2018b). An enriched soil water isotopic composition during the WS and depletion during the DS is comparable to observations made by Hsieh et al. (1998) in an arid to humid transect in Hawaii. Similar differences between depleted winter and enriched summer isotopic profiles in combination with mixing processes were also previously reported (Baram et al., 2013; DePaolo et al., 2004).





In tropical regions, the isotopic composition of precipitation is often correlated with precipitation amount (Araguás-Araguás et al., 2000), and this temporal variation is critical for pore water stable isotope studies, especially during the WS. Taking the variation of vapor source (from precipitation) into account and by comparing the isotopic composition of soil water with the original water input, we can estimate the fraction of evaporation loss for an isotope mass balance.

## 4.2 Fraction of evaporation estimation

Kinetic fractionation in the shallow soil is relatively small in tropical climates. Our observations point to kinetic fractionation down to a depth of ~0.2 m, shallower than the average depth in temperate regions (~0.3 m) (Gazis and Feng, 2004; Sutanto et al., 2012), the Mediterranean (~0.5 m) (Oshun et al., 2016; Simonin et al., 2014), or in arid climates (~3 m) (Allison and Hughes, 1983; Singleton et al., 2004). Shallow soils exhibit a decreasing trend of $F_E$ from the beginning of plant growth towards the end from fields in DS (Fig. 5). Pore water in rice fields has low $F_E$ in deep soils, and especially below 0.4 m when reaching the end of the DS, while in maize fields, it was small below ~0.2 m. Under a controlled laboratory experiment on evaporating soil columns, Rothfuss et al. (2010) observed higher pore evaporation fractionation in the top 0.2 m of soil, which diminished below 0.4 m in loamy soil for deep-rooted perennial grass. During the WS, $F_E$ was higher in the shallow soil (due to more pronounced kinetic fractionation processes compared to the DS) and decreased towards the end of the growth period. In a laboratory experiment by Rothfuss et al. (2010), comparable observations were found where $F_E$ changed over time, with 100 % from bare soil that decreased from 94 % to 5 % with respect to the time (from 16 to 43 days after the seeding) of perennial grass. However, the $F_E$ during the WS can be biased due to (I) high variability of isotopic composition during intense precipitation events, (II) effects related to the formation of hydrogen compounds (described in section 4.3), and (III) higher crop evapotranspiration than the reference evapotranspiration.

The values we obtained refer to the fraction of water loss from the matrix and small/intermediate pores. We must take into account that macropore components cannot be determined with this method. Using the CROPWAT model (FAO 2009) forced with meteorological data for Los Baños, Philippines, we estimated an annual average reference evapotranspiration rate of 3.65 mm d$^{-1}$, with a DS average of 3.96 mm d$^{-1}$ and a WS average of 3.33 mm d$^{-1}$. From transplanting to harvest, crop evapotranspiration increased from around 2.4 to 5.0 mm d$^{-1}$ during the DS, and of 3.4 to 4.1 mm d$^{-1}$ during the WS, respectively. This is in the range with other published evapotranspiration rates. For example, daily evapotranspiration rates of 3.74–3.90 mm d$^{-1}$ from maize and 4.13–4.36 mm d$^{-1}$ from rice are given by Alberto et al. (2014) for the same study site during the DS. Furthermore, in the tropics, evapotranspiration rates of 6–7 mm d$^{-1}$ during the DS and 4–5 mm d$^{-1}$ during the WS were reported (Datta, 1981). By a simple calculation, we derived approximate evaporation of ~50–80 % from effective precipitation plus irrigation for the entire year. Values of about 30 % evaporation were reported for Asia (Bouman et al., 2005), and 40 % for floodwater in temperate Australia (Simpson et al., 1992). However, Wei et al. (2018) showed that an isotopic approach can also lead to higher estimates of the fractions compared to model results for rice and maize in Tsukuba, Japan. Overall, we conclude that the isotope method provides comparable results to previous studies.





### 4.3 Fractionation differences between $\delta^2H$ and $\delta^{18}O$ and uncertainties

Apart from the highly depleted isotopic signal for $\delta^2H$ observed in deep soil under wet rice fields during the WS (Fig. 3b), there was a systematic deviation of about 20 % between $\delta^2H$ and $\delta^{18}O$ fractionation at shallow soil and 40 % at deep soil. (Fig. 5a–c). This may have resulted from the formation of different hydrogen compounds under continuous inundation conditions.

Flooding affects soils chemically, physically, and biologically, resulting in a reduction of redox potential (Fageria et al., 2011; Zhang et al., 2015). Due to the anaerobic conditions that developed in the soil, hydrogen compounds such as $CH_4$, $H_2S$, $H_2$, and $NH_4^+$ can be produced via microbial anaerobic respiration (Fageria et al., 2011; Gerardi, 2003). Formation of these hydrogen compounds leads to isotopic exchange and bias in $\delta^2H$, as observed by Baram et al. (2013) in clay soils below ponded wastewater conditions. $CH_4$ emissions in wet rice fields on our study site were higher during the WS compared to the DS

(Weller et al., 2016), and this may have caused lower slopes in the dual isotope plots as observed (Fig. 4a–c).

Furthermore, the equilibrium constant for isotopic partitioning of liquid water with vapor ($1000\ln\alpha$) is a function of the temperature (here we present the values at 27°C) and the sign of the value (positive), e.g., $H_2O_{(l)} \leftrightarrow H_2O_{(g)}$ for $\delta^{18}O$ +9.2 (Freidman and O'Neil, 1977; Majoube, 1971) and +74.3 for $\delta^2H$ (Majoube, 1971). Water vapor $\delta^2H$ further isotopically fractionates with $CH_{4(g)}$ ($1000\ln\alpha$=+23.4, see Bottinga, 1969), $H_2S_{(g)}$ ($1000\ln\alpha$=+851.0 as in Galley et al., 1972; Clark and

Fritz, 1997), as well as liquid water with $CH_{4(g)}$ with $1000\ln\alpha$=+242.1 (Horibe and Craig, 1995), leading to higher $\delta^2H$ (enriched) in both phases. Moreover, liquid water and water vapor further manifest an equilibrium with $H_{2(g)}$ with higher equilibrium fractionation (Bottinga, 1969; Rolston et al., 1976). As a result, the assumption of $\delta^2H$ enrichment is further reinforced. The difference between $\delta^2H$ and $\delta^{18}O$ has been found to be more pronounced at a greater depth, stipulating formation of hydrogen compounds in deeper soil (Fig. 5 a–c). Besides, exchange rates and fractionation with kaolinite and

smectite (Gilg and Sheppard, 1996) are faster and more pronounced for $\delta^2H$. The assumption for this dissimilarity between $\delta^2H$ and $\delta^{18}O$ can be quantified by a sensitivity analysis, giving a relative depletion by 5±2 ‰ of $\delta^2H$. Because of the above processes, bias can result in the calculation of $F_E$ during the WS. Due to the high standard deviation of the isotopic composition in extreme precipitation events during the WS, prediction of the original water source at a time was also more uncertain. The $F_E$ values are sensitive to the isotopic composition of atmospheric vapor and original water input, nevertheless, only seasonal

averages were assigned in the calculation. This difference was not prominent in wet rice fields during the DS, where oxidizing conditions occurred in time gaps between irrigation events; it was also not observed in dry rice and maize fields.

In addition, vacuum-extracted soil water also contains bound water plus adsorbed water, making isotopic composition lower (Gaj et al., 2017; Velde, 2012), separate from additional systematic errors resulting from the extraction method (Orlowski et al., 2016). High water-holding capacity (Brouwer et al., 2001; Hazelton and Murphy, 2016) and the shrinking and swelling

behavior (Baram et al., 2013; Dasog et al., 1988) of clayey soil add complexity to the analysis. Determination of $\alpha_k$ can also result in estimations errors of 1 to 29 %, depending on the value of $\alpha_k$ and the day of the partition (Rothfuss et al., 2010).




## 5 Conclusions

We identified water flow dynamics in the field, controlled by three main processes: (I) in non-flooded conditions, the isotopic enrichment produced at the soil surface moves downwards while there is an upwards evaporative flux resulting in an exponential profile; (II) in flooded conditions, the isotopic enrichment of surface water caused by evaporation is reflected in
the surface soil, based on a piston-flow-type movement from the surface ponded water, therefore, the explanation of wet rice isotopic profiles is more complex; and (III) in dry soils, especially in maize, there is a preferential flow through cracks in addition to matrix flow.

We identified four main processes, which may be responsible for variations in the natural isotopic profile: physical soil evaporation, soil water movement, redistribution by roots and transpiration, and the refilling of deep soil through preferential
flows via desiccation cracks. This leads to the conclusion that isotopic profiles develop via diffusion processes in the shallow soil and are then transported by advection in the matrix or in macropores or cracks. During flooding, the signal at the surface is reset by infiltration, redistributed in the soil profile, and subsequently smoothened by the root system and transpiration. Evapotranspiration decreases the soil moisture but preserves the profile.

There was a clear isotopic separation between shallow and deep soil, with higher enrichment in shallow soil. Deep soil in wet
rice fields often presented inverted evaporated profiles because of lower compartments carrying over the history of the transported evaporation signal from the previous season. Shallow soils in maize fields showed a stronger soil evaporation effect than rice fields. However, compared to the original water input, greater water loss was estimated during the WS compared to the DS when referring to evaporation from the soil matrix (supported also by higher lc-excess values). Soil evaporation in wet rice during the WS was largely obscured by short-term variability of high precipitation events. Reduction
processes under anaerobic conditions may have affected $\delta^2H$ and caused relatively depleted $\delta^2H$ values compared to $\delta^{18}O$. Therefore, a higher difference between $\delta^2H$ and $\delta^{18}O$ in liquid and vapor phases was found in wet rice fields during the WS due to the equilibration of $\delta^2H$ with hydrogen compounds. This study suggests that this is a common effect in flooded rice fields affecting stable isotope studies by causing a bias due to the compounds formed in reducing environments.

With our method, we can determine flow processes, unproductive soil water losses and relate redistribution patterns to crop
diversification and seasonal differences. However, another independent tool is needed to calculate total evapotranspiration for validation such as eddy covariance, CROPWAT model. In conclusion, our hypothesis of reducing the unproductive water losses by introducing dry seasonal crops is supported by isotope data.

*Author contributions*. A.M.; Field sampling and laboratory analysis, data visualization, original draft preparation, A.M., C.K.,
L.B.; data interpretation, C.K., P.K., L.B.; review and editing.

*Acknowledgments.* This research was undertaken as part of the ICON project phase II (Introducing Non-Flooded Crops in Rice-Dominated Landscapes: Impact on Carbon, Nitrogen and Water Cycles) under the sub-project 07 (Monitoring and modeling of water and water-related nutrient fluxes in rice-based cropping systems) funded by the DFG Research Unit



FOR1701, BR2238/9-2. We acknowledge with gratitude the International Rice Research Institute, the Philippines, and Reiner Wassmann for providing research, space, and support. We gratefully acknowledge technical support by Heathcliff Racela during the experiments. We would like to thank Samantha Serratore for editing the final manuscript.

*Competing interests*. The authors declare that they have no conflict of interest.

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

30





**Tables and Figures**

**Table 1.** Soil texture and average bulk densities of different depths along the soil profile

| Soil depth (m) | Texture | | | Bulk density (g cm$^{-3}$) | |
|---|---|---|---|---|---|
| | Clay (%) | Silt (%) | Sand (%) | Rice fields | Maize fields |
| 0.0–0.1 | 58.3 | 33.4 | 8.4 | 0.92±0.03 | 1.17±0.02 |
| 0.1–0.2 | 59.5 | 30.9 | 9.7 | 1.02±0.03 | 1.13±0.04 |
| 0.2–0.4 | 58.9 | 29.6 | 11.5 | n.a | n.a |
| 0.4–0.6 | 50.0 | 26.7 | 23.4 | n.a | n.a |

10 **Table 2.** Mean±standard deviation (SD) of all water samples (rainwater weighted mean (RW), irrigation water (IW), groundwater (GW), and surface water (SW)) from different crops (wet rice, dry rice, and maize) during the wet season (WS) and dry season (DS).

| Season | Crop | Water type | $\delta^2$H±SD ‰ | $\delta^{18}$O±SD ‰ |
|---|---|---|---|---|
| WS | | RW | -26.82±2.30 | -4.42±0.34 |
| | | IW | -32.00±3.25 | -4.34±0.65 |
| | Wet rice | GW | -23.76±5.24 | -3.03±1.21 |
| | Wet rice | SW | -24.06±7.36 | -3.22±1.69 |
| DS | | RW | 8.73±0.62 | 0.05±0.08 |
| | | IW | -34.60±3.56 | -4.89±0.56 |
| | Wet rice | GW | -14.66±7.46 | -1.75±1.27 |
| | Wet rice | SW | -14.15±9.41 | -1.80±1.41 |
| | Dry rice | GW | -12.56±8.75 | -1.37±1.52 |
| | Maize | GW | -22.57±7.60 | -3.10±1.19 |



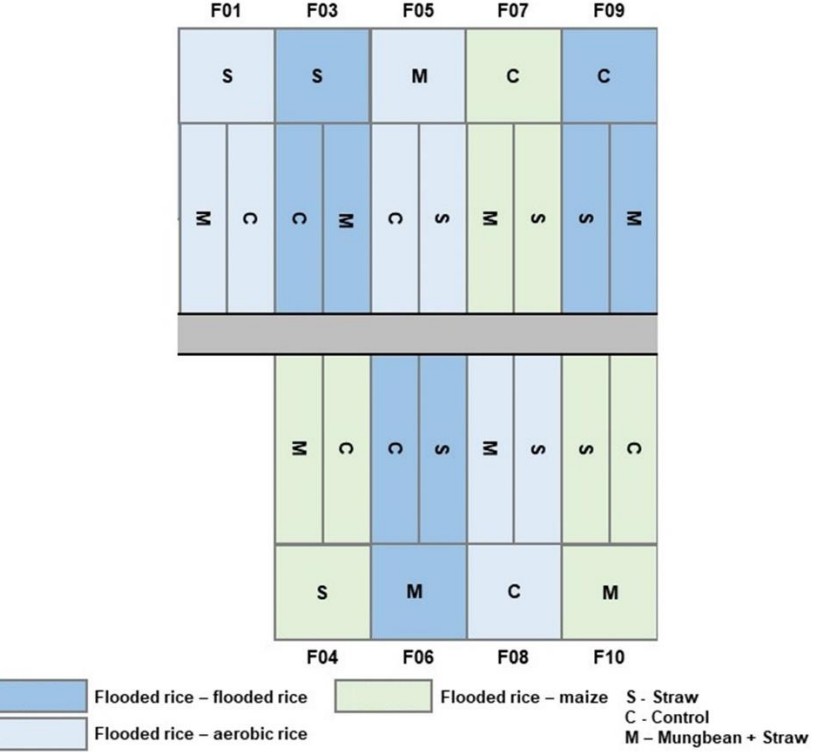

Figure 1. Experimental field design. The experiment consisted of nine fields (F) with three different water management practices. During the wet season, all fields were cultivated with wet rice, while during the dry season, three fields each were cultivated with wet rice, dry rice, and maize. Each field is divided into three different treatments (S=straw incorporated in the soil, C=control, M=straw plus mung bean as an inter-crop in the dry to wet transition period).

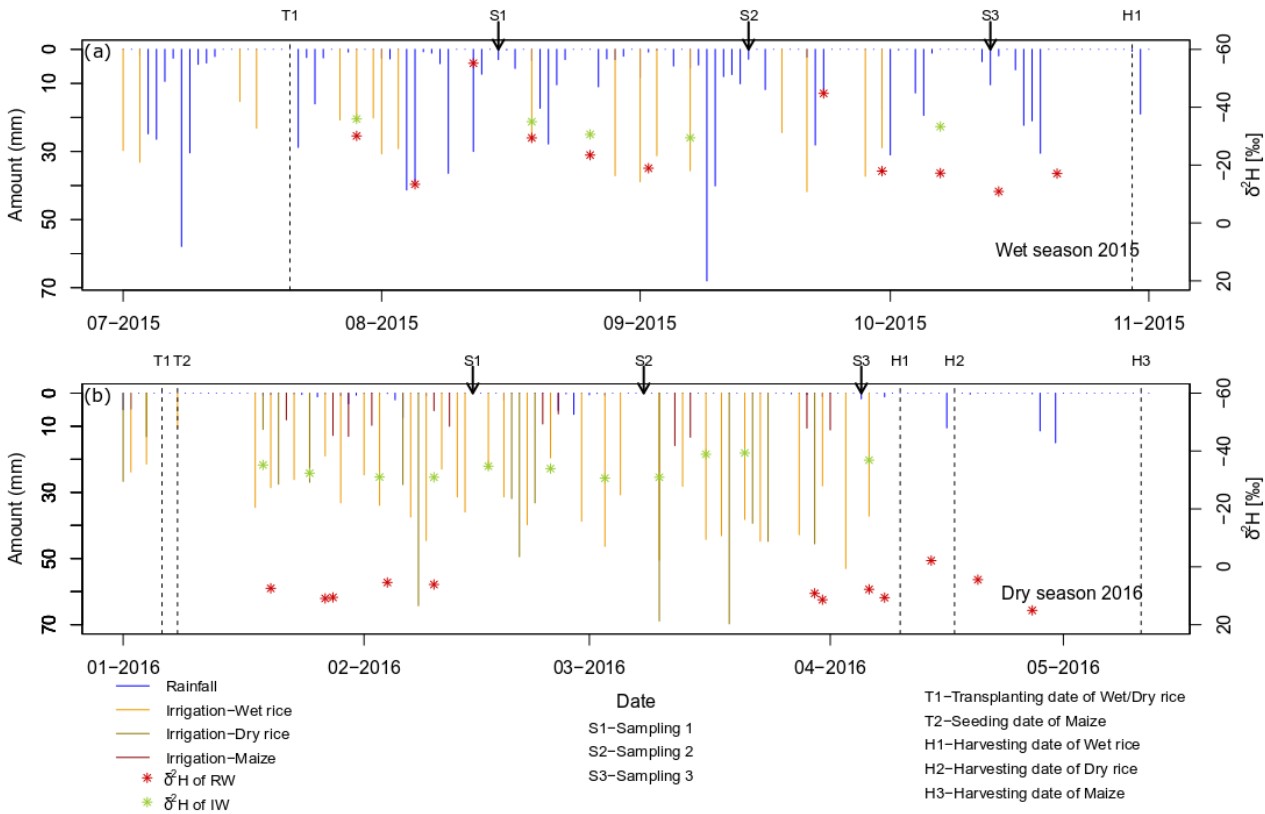

Figure 2. Temporal variation of water inputs (rainfall and irrigation water) of wet rice, dry rice and maize fields for the wet season 2015 (top) and dry season 2016 (bottom). Three main sampling dates during each season together with transplanting and harvesting dates are marked. Values of $\delta^2$H are presented for rainwater (RW) and irrigation water (IW) during both seasons.



Figure 3. Depth profiles of means +/- standard deviation for $\delta^{18}O$ /‰, $\delta^{2}H$ /‰, soil water content (SWC) [%], and lc-excess /‰ from three main growing stages (GS1 to GS3) of wet rice (a–d) during the wet season (WS), and wet rice (e–h), dry rice (i–l), maize (m–p) during the dry season (DS). Seasonal averages +/- standard deviation of all the water sources (rainwater (RW), irrigation water (IW), groundwater (GW) and surface water (SW) isotopic values are displayed at the top and bottom of the soil profiles.





Figure 4. Dual ($\delta^{18}$O, $\delta^2$H) isotope plots of soil water at 0–0.6 m depth, and ranges of other water sources (rainwater, irrigation water) from growing stage GS1 (a, d, g, j), GS2 (b, e, h, k), and GS3 (c, f, i, l), from wet rice (a–c) during the wet season (WS), and wet rice (d–f), dry rice (g–i) as well as maize (j–l) during the dry season (DS) in comparison to the local meteoric water line (LMWL) and global meteoric water line (GMWL). The gray shaded areas represent the 95 % confidence interval of the linear regression lines.





Figure 5. The fraction of evaporation loss ($F_E$) (Eq.1) following $\delta^{18}O$, $\delta^2H$ from three main growing stages: growing stage GS1 (a, d, g, j), GS2 (b, e, h, k) and GS3 (c, f, i, l) of wet rice (a–c) during the wet season (WS), wet rice (d–f), dry rice (g–i), maize (j–l) during the dry season (DS). Mean values at each depth (0–0.6 m) are displayed with +/- standard deviations.