# Peer review of "Investigating unproductive water losses from irrigated agricultural crops in the humid tropics through analyses of stable isotopes of water"

_Hydrology and Earth System Sciences, 2019_

## Referee Comment (RC1) · Matthias Beyer (Referee) · 20 Jul 2019

In their manuscript 'Estimating water flux and evaporation losses using stable isotopes of soil water from irrigated agricultural crops in tropical humid regions' (hess-2019-213), Mahindawansha et al. investigate the effect of different crop rotations (wet rice/dry rice/maize) in seasonally flooded/ irrigated rice fields. The authors quantified the fraction of soil water evaporation in irrigated agricultural fields while also taking into account the effect of crop species and various growing stages using the Craig-Gordon model.

The topic of the study is interesting and timely but, in brief, I have mixed feelings on the manuscript. While it is clearly visible that the collected dataset can be valuable

for addressing the objectives of the study, there are several points that need to be addressed in order to make this contribution really valuable for the reader. First, I have the feeling that the manuscript is lacking some internal review before publishing. The grammar is partially very poor, and I feel that several aspects (e.g. clear statement of the objectives and focus on those in the results/discussion section) should have clarified before submission. I started correcting/improving the grammar, but gave up fast on it because it became clear that major efforts are needed which I as reviewer cannot provide. Second, the combined effect of Transpiration and Evaporation should be much better addressed throughout the manuscript. Recent studies proved that transpiration is generally a much greater flux compared to Evaporation, and in a study like the presented those two need to be looked at conjunctively. In that regard, also the title is confusing, because when reading 'estimation of water fluxes', one would actually expect a water balance for the different systems, but effectively the only flux quantified is evaporation. In addition, I was confused multiple times because I was not sure if the authors speak about evaporation or evapotranspiration? (see later comments). Also, I was wondering multiple times if the authors really refer to soil evaporation when speaking of wet rice? If the field is flooded, it would be more open water evaporation? Having that said, I cannot recommend publishing this manuscript as is. Though the topic and study are interesting and have great potential, this is often not fully explored. With more precisely stated objectives and a subsequent focus on addressing those, I encourage the authors to improve the manuscript and increase the quality and impact of the publication. I wish the authors good luck with the revision of the manuscript. Kind regards, Matthias Beyer Down below, further detailed comments can be found. The authors state: 'None of the studies conducted so far have quantified the fraction of soil water evaporation in irrigated agricultural fields while also taking into account the effect of crop species and various growing stages. Does it make sense to calculate the evaporation from soils for wet rice, which is cultivated in a flooded system (as the authors state) → evaporation would be from open water surface anyways Suggested objective: study the effect of crop species and various growing stages on evaporation

in rotation systems

Title: why first singular (flux) and then plural (losses)? - The abstract needs to be improved. There are many sloppy formulations and bad grammar. The results section of the abstract should be underpinned with numbers. What are the implications of this study and how does it help to improve management or our understanding of such systems? - How to compare an irrigated/flooded rice field with a field under natural conditions in terms of water isotope interpretations? - While reading the introduction, I wonder if the authors solely mean soil evaporation when they use the wording "evaporation" or if they actually mean "Evapotranspiration" (sometimes, evaporation is used for ET). The authors state that they are interested in studying soil evaporation, but can you look at one (E) without the other (T) in a combined system? - Methods - Extraction at 200 degrees Celsius. . .good, because very clay-rich. . .but. . .was organic contamination checked? (upper soil layers and plants) - Craig and Gordon modeling part should be written more concise. What is the difference between the isotopic signal of the soil and the original isotopic signal of soil water? (do the authors mean the 'initial signal ofter rain/irrigation?). How justified are the assumptions made (and those are many)? For the results, it would be interesting to see if the fraction calculation fits with the modelled results

P1.l. 13 advance better: improve P1.l.18: progressed through the growth – bad grammar P1.l.23 compared to over P2 l.6-11: not only in recent years, this has been studied since Allison et al. in the 80's. . .has not been studied as much compared to what? p.2.l.13 it p.2.l.20-32: this is well-written! p.3.l.5: Our objectives during this study are the objectives of this study are p.3. l. 5-8: Objectives should be formulated clear and concise p.3. l. 21: constancy consistency p.3. l. 20-23: if the mung bean plot was not used it is not necessary to mention it here p.4. l. 6: the model controls?...grammar p.4. l. 8: mixing from macropore flow from cracks?...grammar p.6. l. 25/26: the shape of the isotopic profiles in the shallow soil water changed depending on the crop and growth stage. → only because of that or also other factors – irrigation water isotope

values, precipitation, radiation? What are the conclusions of the authors regarding the magnitudes of evaporation? (Are these numbers given as fraction of total evapotranspiration?) Fig. 3: bad resolution Section 4.3.: the statements here are very interesting and it is appreciable that the authors introduce this discussion. unfortunately they question parts of the isotopic data presented in the study Conclusion: - throughout the manuscript, the phrase 'redistribution via plants/roots/etc. appears frequently', but it is not discussed anywhere. I suggest leaving this out or providing further evidence. - 'the conclusion that isotopic profiles develop via diffusion processes in the shallow soil and are then transported by advection in the matrix or in macropores or cracks' → please rephrase, poor grammar p.13., l. 13: Do the authors really mean Evapotranspiration or rather Transpiration?

Please use continued page-numbering for revised version

———————————————

In their manuscript 'Estimating water flux and evaporation losses using stable isotopes of soil water from irrigated agricultural crops in tropical humid regions' (hess-2019-213), Mahindawansha et al. investigate the effect of different crop rotations (wet rice/dry rice/maize) in seasonally flooded/ irrigated rice fields. The authors quantified the fraction of soil water evaporation in irrigated agricultural fields while also taking into account the effect of crop species and various growing stages using the Craig-Gordon model.

The topic of the study is interesting and timely but, in brief, I have mixed feelings on the manuscript. While it is clearly visible that the collected dataset can be valuable for addressing the objectives of the study, there are several points that need to be addressed in order to make this contribution really valuable for the reader.

First, I have the feeling that the manuscript is lacking some internal review before publishing. The grammar is partially very poor, and I feel that several aspects (e.g. clear statement of the objectives and focus on those in the results/discussion section) should have clarified before submission. I started correcting/improving the grammar, but gave up fast on it because it became clear that major efforts are needed which I as reviewer cannot provide.

Second, the combined effect of Transpiration and Evaporation should be much better addressed throughout the manuscript. Recent studies proved that transpiration is generally a much greater flux compared to Evaporation, and in a study like the presented those two need to be looked at conjunctively. In that regard, also the title is confusing, because when reading 'estimation of water fluxes', one would actually expect a water balance for the different systems, but effectively the only flux quantified is evaporation. In addition, I was confused multiple times because I was not sure if the authors speak about evaporation or evapotranspiration? (see later comments). Also, I was wondering multiple times if the authors really refer to soil evaporation when speaking of wet rice? If the field is flooded, it would be more open water evaporation?

Having that said, I cannot recommend publishing this manuscript as is. Though the topic and study are interesting and have great potential, this is often not fully explored. With more precisely stated objectives and a subsequent focus on addressing those, I encourage the authors to improve the manuscript and increase the quality and impact of the publication. I wish the authors good luck with the revision of the manuscript.

Kind regards,

Matthias Beyer

Down below, further detailed comments can be found.

The authors state: 'None of the studies conducted so far have quantified the fraction of soil water evaporation in irrigated agricultural fields while also taking into account the effect of crop species and various growing stages. Does it make sense to calculate the evaporation from soils for wet rice, which is cultivated in a flooded system (as the authors state) → evaporation would be from open water surface anyways

Suggested objective: study the effect of crop species and various growing stages on evaporation in rotation systems

**Fig. 1.**

---

## Referee Comment (RC2) · Anonymous Referee #2 · 30 Jul 2019

Review of Hydrology and Earth System Sciences Manuscript: hess-2019-213 Title: Estimating water flux and evaporation losses using stable isotopes of soil water from irrigated agricultural crops in tropical humid regions Authors: Amani Mahindawansha et al.

This manuscript presents seasonal variations in the soil water isotopic profiles and the fraction of evaporation (FE) for different crops (wet rice, dry rice and maize) under flooded and non-flooded irrigation management practices. This topic is interesting for understanding water cycle and water conservation in agricultural fields. However, there are some issues within the manuscript that requires substantial interpretation and

improvement. The following is my detailed comments.

(1) Abstract: Since only FE values were calculated and no water flux of evaporation were determined in this study, the second sentence (P.1,Lines 13-15) should be changed. Other evidences should be given to prove the occurrence of piston type matrix flow or preferential flow besides the isotopic data in the text (P.1,Lines 22-24). It is helpful to supplement important data in the abstract section to clarify the new findings of this study. (2) Introduction: Determination of the soil evaporation flux (E) and the fraction of E in ET (FE) have been widely studied using several methods and techniques for different irrigated crops (Liu et al., 2002; Kool et al., 2014; Sprenger et al., 2016; Zhou et al., 2016). The new scientific merits in this study are not very clearly clarified. (3) Material and Methods: There are straw and non straw applications conducted for different treatments in the experiments (P.3,Lines 20-21). How does the straw application affect the seasonal variations in the FE for different irrigated crops? Please describe in detail how to determine the time when a water shortage occurred in dry rice and maize fields (P.3,Lines 26-27). The gravimetric soil water content is determined traditionally by oven-drying method. Smaller values might be resulted by using the soil water loss in cryogenic water extraction process to determine the soil water content (P.4,Lines 19-20). Root length density was analyzed as described in the P.4, Lines 21-22 in the "Material and Methods" section, but non detailed results were shown in the "3 Results" section. (4) Results: The lc-excess was developed/introduced by Landwehr and Coplen (2006) in respect to River Water Line. They used the lc-excess to determine how the isotopic values of river waters differed from their sources (i.e., precipitation). However, the authors use lc-excess to estimate the deviation in the isotopic values of the soil samples from regional precipitation. I do not find any good argument why the authors use lc-excess since there is no river water sampled during their experiments. The lc-excess is not necessarily needed in this study (P.7, Lines 9-16). Instead, the deviation of soil isotopic values from LMWL/GMWL is already indicating the evaporation process and it is more commonly used method. Lower <delta> indicates condensation process and higher <delta> indicates evaporation process. (5) Discussion: The authors

estimate the annual average reference evapotranspiration rates in dry season and wet season, respectively. Does the "evaporation of ∼50-80%" in P.11, Line 28 mean evapotranspiration? What is the difference between evapotranspiration and evaporation in this study? (6) Conclusion: Seasonal distribution of soil water content and isotopic profiles was analyzed in this study, but no fluxes of unproductive soil water losses were found. Therefore, the sentence in P.13, Lines 24-25 is required to be reorganized. (7) The English writing of this manuscript should be polished further. There were some grammar errors in this paper and some sentences were confusing.

References: [1] Liu, C.M., Zhang, X.Y., Zhang, Y.Q., 2002. Determination of daily evaporation and evapotranspiration of winter wheat and maize by large-scale weighing lysimeter and micro-lysimeter. Agr. For. Meteorol. 111, 109-120. [2] Kool, D., Agam, N., Lazarovitch, N., Heitman, J. L., Sauer, T. J., Ben-Gal, A., 2014. A review of approaches for evapotranspiration partitioning. Agric. For. Meteor. 184, 56-70. [3] Zhou, S., Yu, B.F., Zhang, Y., Huang, Y.F., Wang, G.Q., 2016. Partitioning evapotranspiration based on the concept of underlying water use efficiency. Water Resour. Res. 52, 1160-1175. [4] Sprenger, M., Leistert, H., Gimbel, K., Weiler, M., 2016. Illuminating hydrological processes at the soil-vegetation-atmosphere interface with water stable isotopes. Rev. Geophys. 54, doi:10.1002/2015RG000515.

---

## Author Comment (AC1) · 10 Aug 2019

Dear Reviewer,

Thank you very much for your valuable comments on our manuscript entitled, "Estimating water flux and evaporation losses using stable isotopes of soil water from irrigated agricultural crops in tropical humid regions" that will help us to further improve the manuscript.

We agree that the distinction of physical evaporation processes and transpiration is very important for the interpretation of isotope profiles. We will review the wording and

terminology of the text very carefully to make this distinction always very clear for the reader. We will also edit the title and abstract correspondingly. As proposed, we will send the manuscript to a proofreading service for English correction (despite that the initial manuscript had already been checked by a native proofreader). Soil water samples were checked for organic contaminants and were found not to be contaminated. This will be explained in the revised version of the paper in more detail. We already submitted high-resolution graphs separately in the first submission, but obviously they have not been included in the PDF. We hope they will make it into the next version.

---

## Author Comment (AC2) · 10 Aug 2019

Dear Reviewer,

Thank you very much for your valuable comments on our manuscript entitled, "Estimating water flux and evaporation losses using stable isotopes of soil water from irrigated agricultural crops in tropical humid regions" that will help us to further improve the manuscript.

We will address your comments and we will then send the manuscript again to a proofreading service for English correction (despite that the initial manuscript had already

been checked by a native proofreader). We will improve the abstract and introduction in order to highlight the novelty and clarify the objectives of the work. In the next version of our manuscript, we will particularly address the use of lc-excess and why it has advantages over the GMWL based d-excess.
* * *

---

## Author Comment (AC3) · 27 Sep 2019

Dear Matthias Beyer,

We would like to thank you for the valuable feedback provided for our manuscript entitled, "Estimating water flux and evaporation losses using stable isotopes of soil water from irrigated agricultural crops in tropical humid regions". Your comments were very helpful to improve the manuscript. Please find our point-by-point responses (starting with a '#') to the comments below.

We believe that the modifications based on the referees' comments have resulted in an

[Figure]

improved manuscript and hope that it is now suitable for consideration for publication as a research paper in Hydrology and Earth System Sciences.

We look forward to hearing from you.

Best regards, On behalf of the authors, Amani Mahindawansha
**As proposed, we will carry out an internal review again and work on the English language. A native proofreader had already checked the initial manuscript, and we were slightly puzzled when reading that the version submitted was still flawed. Nevertheless, we will send the paper again for proofreading.**
Second, the combined effect of Transpiration and Evaporation should be much better addressed throughout the manuscript. Recent studies proved that transpiration is generally a much greater flux compared to Evaporation, and in a study like the presented those two need to be looked at conjunctively. In that regard, also the title is confusing, because when reading 'estimation of water fluxes', one would actually expect a water balance for the different systems, but effectively the only flux quantified is evaporation. In addition, I was confused multiple times because I was not sure if the authors speak about evaporation or evapotranspiration? (see later comments).

**We agree that transpiration is larger than evaporation and that for a water balance analysis. However, in our work, we look at evaporation as an unproductive loss term of the water balance. We will revise the title accordingly ('Investigating unproductive water losses from irrigated agricultural crops in the humid tropics through analyses of stable isotopes of water'). Hence, we have only investigated soil evaporation and the water losses from the soil due to the evaporation process. In the revised version, we will correct the wording and terminology carefully to make the distinction of evaporation and evapotranspiration always very clear for the reader.**

Also, I was wondering multiple times if the authors really refer to soil evaporation when speaking of wet rice? If the field is flooded, it would be more open water evaporation?

**Yes, we agree that in a temporarily flooded field the evaporation occurs from the open ponded water body during the time of flooding. During ponding, infiltration modifies the soil water isotopic composition in the uppermost part of the profile and re-evaporation of infiltrated water has been interpreted and termed as soil evaporation. We will make the distinction between open water infiltration during ponding and soil evaporation clear.**

Having that said, I cannot recommend publishing this manuscript as is. Though the topic and study are interesting and have great potential, this is often not fully explored. With more precisely stated objectives and a subsequent focus on addressing those, I encourage the authors to improve the manuscript and increase the quality and impact

of the publication. I wish the authors good luck with the revision of the manuscript.

**The revised version of the manuscript will focus on unproductive evaporation losses. We will work on consistent use of the terminology, state our objective clearly and streamline the discussion towards it. After this revision, we hope the significance and novelty of our study can be conveyed.**

Down below, further detailed comments can be found. The authors state: 'None of the studies conducted so far have quantified the fraction of soil water evaporation in irrigated agricultural fields while also taking into account the effect of crop species and various growing stages. Does it make sense to calculate the evaporation from soils for wet rice, which is cultivated in a flooded system (as the authors state) → evaporation would be from open water surface anyways Suggested objective: study the effect of crop species and various growing stages on evaporation in rotation systems

**Please see our reply to the terminology of soil evaporation vs. evaporation from open water bodies, which we will make clearer in the revised version of the manuscript. Further, we followed the suggestion and specified our objectives. 1) Title: why first singular (flux) and then plural (losses)?**

**We revised the title to 'Investigating unproductive water losses from irrigated agricultural crops in the humid tropics through analyses of stable isotopes of water'**

2) The abstract needs to be improved. There are many sloppy formulations and bad grammar. The results section of the abstract should be underpinned with numbers.

**We revised the abstract and added numbers to the results section.**

3) What are the implications of this study and how does it help to improve management or our understanding of such systems? How to compare an irrigated/flooded rice field with a field under natural conditions in terms of water isotope interpretations?

**We thank the reviewer to make these points and we added the following section to the discussion with regard to the first part of the comment. "Water losses via soil evap-**

[Figure]

oration is a major unproductive loss term of the water budget, especially during early growing stages. To improve water management, a more detailed understanding of water cycles of rice-based cropping systems is required. Apart from reducing leaching losses, the second most important water-saving measure in these systems is reducing soil evaporation. Therefore, our study helps to increase understanding of soil water transport processes and evaporation losses from soil in response to crop rotation systems. Farmers should apply mitigation methods to reduce soil water evaporation. e.g. by mulching, or growing plant cover crops in the fallow period."

With regard to the second question: We compared flooded and non-flooded fields. In this sense, the dry rice and maize fields are representing natural conditions.

4) While reading the introduction, I wonder if the authors solely mean soil evaporation when they use the wording "evaporation" or if they actually mean "Evapotranspiration" (sometimes, evaporation is used for ET). The authors state that they are interested in studying soil evaporation, but can you look at one (E) without the other (T) in a combined system?

**The terminology for evaporation and evapotranspiration were carefully checked and addressed avoiding the confusions. Our work is focusing on unproductive water losses as explained above. Hence, transpiration is not part of this study. We additionally added the following sentence into the discussion to make it clearer. "With isotope methods, we only estimated unproductive evaporation losses from soil, because transpiration does not change the isotopic signal as it is known as a non-fractionating process (Zimmermann U. et al., 1967). Therefore, a different tool was used to estimate evapotranspiration. We rigorously tested our results and checked their plausibility by reviewing regional data reported in the literature, and by using the CROPWAT modeling approach."**

5) Methods - Extraction at 200 degrees Celsius. . .good, because very clay-rich. . .but. . .was organic contamination checked? (upper soil layers and plants)

**Isotopic composition data of all water source types were checked for spectral interferences using the Spectral Contamination Identifier (LWIA-SCI) post-processing software (Los Gatos Research Inc.). None of the soil water samples was contaminated. This will be explained in the revised version of the paper in more detail.**

6) Craig and Gordon modeling part should be written more concise.

**We will revise section 2.4 and present the Craig-Gordon model in a more concise way.**

7) What is the difference between the isotopic signal of the soil and the original isotopic signal of soil water? (do the authors mean the 'initial signal after rain/irrigation?). How justified are the assumptions made (and those are many)?

**The original water signal refers to the initial isotopic composition of input sources to the soil water. During the wet season, larger precipitation events replenish the soil water store. During the dry season, irrigation is the main water input method due to small and rare precipitation events. Therefore, we took the amount-weighted mean values of precipitation for the wet season and or irrigation water for the dry season as initial signals. We edited the sentence as follows to make it clear. "The original isotopic signal, $\delta$p, is the initial water added to the soil (input water). During the WS, $\delta$p was estimated as the weighted average isotopic signal from most frequent large precipitation events, and as the weighted mean of the irrigation water during the DS."**

8) For the results, it would be interesting to see if the fraction calculation fits with the modelled results

**Our approach offers the advantage of identifying unproductive evaporation from open water or soil, the water balance model provides data on potential evaporation and actual evaporation calculated from soil moisture constraints. These methods are complementary as they provide different fractions of the water balance. Still, as the soil water balance method does not provide information on the amount of unproductive losses,**

respective fractionation cannot be calculated.

9) P1.l. 13 advance better: improve

**Edited as recommended**

10) P1.l.18: progressed through the growth – bad grammar

**Corrected.**

11) P1.l.23 compared to over

**Edited as recommended**

12) P2 l.6-11: not only in recent years, this has been studied since Allison et al. in the 80's. . .has not been studied as much compared to what?

**We will delete the last sentence in this section, as it does not provide any further insight.**

13) p.2.l.13 it p.2.l.20-32: this is well-written!

**Thank you!!!**

14) p.3.l.5: Our objectives during this study are the objectives of this study are

**Edited as recommended.**

15) p.3. l. 5-8: Objectives should be formulated clear and concise

**We followed the reviewer suggestion and improved the way of presenting the objectives.**

16) p.3. l. 21: constancy consistency

**Corrected as recommended.**

17) p.3. l. 20-23: if the mung bean plot was not used it is not necessary to mention it here

**This sentence has been deleted. We will include a note in the figure caption of Fig. 1 why mung bean plots are depicted in the figure.**

18) p.4. l. 6: the model controls?...grammar

**Corrected.**

19) p.4. l. 8: mixing from macropore flow from cracks?...grammar

**Corrected.**

20) p.6. l. 25/26: the shape of the isotopic profiles in the shallow soil water changed depending on the crop and growth stage. → only because of that or also other factors – irrigation water isotope values, precipitation, radiation?

**We agree that this statement was incomplete and hence deleted it. In the revised version of the manuscript, we will streamline the paper according to the main objectives.**

21) What are the conclusions of the authors regarding the magnitudes of evaporation? (Are these numbers given as fraction of total evapotranspiration?)

**Yes, the fractions of evaporation are given as percentages. The magnitude of evaporation is higher at the beginning of the growing period and decreases towards the end of the season. We revised the section accordingly and explain this in more detail and discuss our results in relation to other published values.**

22) Fig. 3: bad resolution

**We already submitted high-resolution graphs separately in the first submission, but obviously, they have not been included in the PDF. We hope it will work out this time.**

23) Section 4.3.: the statements here are very interesting and it is appreciable that the authors introduce this discussion. unfortunately they question parts of the isotopic data presented in the study

\# We found this observation is interesting of formation of different hydrogen compounds under continuous inundation conditions. We think that the limits and constraints of the method need to be raised and discussed. This effect does not question the study in general as it affects only a limited set of samples. However, we just want to make the reader aware of this issue and we think this has to be considered in future experiments.

24) Conclusion: - throughout the manuscript, the phrase 'redistribution via plants/roots/etc. appears frequently', but it is not discussed anywhere. I suggest leaving this out or providing further evidence. - 'the conclusion that isotopic profiles develop via diffusion processes in the shallow soil and are then transported by advection in the matrix or in macropores or cracks' → please rephrase, poor grammar

\# We will include a new section on this process in the discussion. It will read: "Hydraulic redistribution of water in the vadose zone is an important process of passive transport of soil water along a hydraulic gradient through the rooting system (Richards and Caldwell, 1987). It influences the pore water stable isotopic composition and can reshape the soil water isotopic profile especially in the shallow soil. Sprenger et al. (2016) discussed the significance of hydraulic redistribution for the hydrological process. However, the influence of hydraulic redistribution on the isotopic composition is small (Walter, 2010). In addition, isotopic measurements alone are not sufficient to estimate redistribution volumes (Emerman and Dawson, 1996)." Apart from that, we will revise the grammar in the conclusions.

25) p.13., l. 13: Do the authors really mean Evapotranspiration or rather Transpiration?

\# We mean transpiration fraction of the evapotranspiration and we corrected it in the revised version. We moved that sentence to the section where we explain redistribution patterns in the discussion. As mentioned before, we make sure to carefully maintain the terms and presentation of the wording.

References Emerman, S. H. and Dawson, T. E.: Hydraulic lift and its influence on the water content of the rhizosphere: an example from sugar maple, Acer saccharum, Oecologia, 108(2), 273–278, 1996. Richards, J. H. and Caldwell, M. M.: Hydraulic lift: substantial nocturnal water transport between soil layers by Artemisia tridentata roots, Oecologia, 73(4), 486–489, 1987. Sprenger, M., Leistert, H, Gimbel, K and Weiler, M: Illuminating hydrological processes at the soil‐vegetation‐atmosphere interface with water stable isotopes, Rev. Geophys., 54(3), 674–704, doi:10.1002/2015RG000515, 2016. Walter, K.: Einfluss der Pflanzen auf die Isotopenzusammensetzung des Abflusses in Einzugsgebieten, PhD Thesis, Diplomarbeit, Institut für Hydrologie, Albert-Ludwigs-Universität Freiburg i, 2010. Zimmermann U., MüNnich K. O. and Roether W.: Downward Movement of Soil Moisture Traced by Means of Hydrogen Isotopes, Isot. Tech. Hydrol. Cycle, doi:10.1029/GM011p0028, 1967.

---

## Author Comment (AC4) · 27 Sep 2019

Dear reviewer,

We would like to thank you for the valuable feedback provided for our manuscript entitled, "Estimating water flux and evaporation losses using stable isotopes of soil water from irrigated agricultural crops in tropical humid regions". Your comments were very helpful to improve the manuscript.

Please find our point-by-point responses (starting with a '#') to the comments below.

We believe that the modifications based on the referees' comments have resulted in an

improved manuscript and hope that it is now suitable for consideration for publication as a research paper in Hydrology and Earth System Sciences. We look forward to hearing from you.

Best regards, On behalf of the authors, Amani Mahindawansha
(1) Abstract: Since only FE values were calculated and no water flux of evaporation were determined in this study, the second sentence (P.1,Lines 13-15) should be changed.

\# We agree with the reviewer regarding the water fluxes. In this respect, in the revised version of the manuscript, we changed the title and also the abstract accordingly following the recommendations.

Other evidences should be given to prove the occurrence of piston type matrix flow or preferential flow besides the isotopic data in the text (P.1, Lines 22-24).

\# The results we present here are based on isotopic profiles. From these results, we deduce the respective processes. As this text is part of the abstract, we do not see

how we should add further evidence. However, we had already included this in the discussion together with the evidence for fast mixing of irrigation water and groundwater in maize fields via cracks (Mahindawansha et al., 2018). "In maize fields at our study site, we observed that the groundwater isotopic compositions are strongly influenced by irrigation water suggesting the existence of fast flow conduits (Mahindawansha et al., 2018a). In addition, He et al. (2017) have observed leaching losses of water and nutrients in a lysimeter experiment in the same study site." The pattern of homogeneity and variability of soil moisture can be used as an independent indicator support the interpretation. There is evidence for gravity driven, piston-type matrix flow in the literature based on soil water isotope profiles and chloride concertation (Baram et al., 2013). They also observed accumulation of salts in the deep vadose zone due to fast transport via cracks. We also point this out in the discussion section, where we underline the process understanding and how it was derived.

It is helpful to supplement important data in the abstract section to clarify the new findings of this study.

**We agree with the reviewer and added the main results to the abstract.**

(2) Introduction: Determination of the soil evaporation flux (E) and the fraction of E in ET (FE) have been widely studied using several methods and techniques for different irrigated crops (Liu et al., 2002; Kool et al., 2014; Sprenger et al., 2016; Zhou et al., 2016). The new scientific merits in this study are not very clearly clarified.

**The merit of this study is quantifying the fraction of soil water evaporation in irrigated agricultural fields and taking into account the effect of different crop species and different irrigation and management practice at various growing stages. Additional references listed have been considered in the revised version of the introduction. To better account for previous work, we will include the following sentences: "The determination of soil evaporation and the fraction of evaporation in relation to total evapotranspiration have been widely studied using several methods for different crops. For example, Liu**

et al. (2002) studied evapotranspiration from winter wheat and maize, using weighing lysimeters. Zhou et al. (2016) partitioned evaporation and transpiration fluxes for corn, soya bean, grassland and forests using flux tower measurements. Kool et al. (2014) applied different methods such as chamber, micro-lysimeter, and soil heat pulse to estimate the evaporation and used stable isotopes of water to separate evaporation from transpiration." We do not agree that the research gaps of current studies are not clearly stated. We would like to draw the attention to the end of the introduction where we already stated in first version of the manuscript: "None of the studies conducted so far have quantified the fraction of soil water evaporation in irrigated agricultural fields, while also taking into account the effect of crop species and various growing stages".

(3) Material and Methods: There are straw and non straw applications conducted for different treatments in the experiments (P.3,Lines 20-21). How does the straw application affect the seasonal variations in the FE for different irrigated crops?

**We did not find a significant difference between the isotopic composition of soil water with or without straw application, and therefore pooled the results for each crop. One would have expected a stronger effect of straw application on evaporation. However, the straw was not applied as a typical mulch layer to reduce evaporation, but was partly worked into the soil to reduce crack formation. This has been mentioned in section 2.4. "The isotopic values of the two treatments (straw application and no straw as a control plot) were combined for each crop for further analysis, because there were no significant differences for stable isotopes of water between the fields with the same crop (p>0.05)."**

Please describe in detail how to determine the time when a water shortage occurred in dry rice and maize fields (P.3,Lines 26-27).

**Field workers from the IRRI were responsible for watering dry crops in case of severe soil water shortages. The decision of watering was not set by specific measurements of model application, but by expert knowledge.**

The gravimetric soil water content is determined traditionally by oven-drying method. Smaller values might be resulted by using the soil water loss in cryogenic water extraction process to determine the soil water content (P.4,Lines 19-20).

**We agree that the results of both methods are not directly comparable. We added a sentence to section 2.2 to address this: "The gravimetric soil water content along the soil profiles was determined based on the soil weight loss following cryogenic water extraction. Soil water content determined this way might deviate from the classical oven drying method and result in slightly lower values. However, we use the gravimetric soil water content not as an absolute value, but rather as a relative value to identify differences along the soil profile."**

Root length density was analyzed as described in the P.4, Lines 21-22 in the "Material and Methods" section, but non detailed results were shown in the "3 Results" section.

**We used root length density values to analyze the root growth along the season together with the plant growth. However, we decided to take this part out of the manuscript but forgot to delete it in the respective method's description. Therefore, we also removed these lines in the revised manuscript.**

(4) Results: The lc-excess was developed/introduced by Landwehr and Coplen (2006) in respect to River Water Line. They used the lc-excess to determine how the isotopic values of river waters differed from their sources (i.e., precipitation). However, the authors use lc-excess to estimate the deviation in the isotopic values of the soil samples from regional precipitation. I do not find any good argument why the authors use lc-excess since there is no river water sampled during their experiments. The lc-excess is not necessarily needed in this study (P.7, Lines 9-16). Instead, the deviation of soil isotopic values from LMWL/GMWL is already indicating the evaporation process and it is more commonly used method. Lower indicates condensation process and higher indicates evaporation process.

**Sprenger et al. (2017) stated that "However, we found that lc-excess was advantageous over the deuterium-excess (or single isotope approaches with $\delta 2H$ or $\delta 18O$) for inferring evaporation fractionation, because the lc-excess of the precipitation input is about 0‰ and with relatively little seasonal dynamics, while $\delta 2H$, $\delta 18O$, and d-excess can have an intense seasonal variability." Furthermore, Sprenger et al. (2018) stated that "The lc-excess describes the deviation of a water sample from the LMWL in the dual-isotope plot, which is used to infer soil evaporation processes due to kinetic fractionation of precipitation input" We agree with these arguments. We analyzed soil profiles from different seasons. Therefore, to avoid problems with the seasonality of $\delta 2H$ and $\delta 18O$ or d-excess, we used lc-excess for comparison. We also used lc-excess because Sprenger et al. (Sprenger et al., 2016, 2017, 2018), McCutcheon et al. (2017) have successfully applied it for soil water studies, previously. They specify the advantage of using lc-excess over the d-excess and further explain that water that is experiencing fractionation by evaporation has a negative lc‐excess and plots below the LMWL in a dual‐isotope plot. However, we will address the point more carefully in our reviewed version.**

(5) Discussion: The authors estimate the annual average reference evapotranspiration rates in dry season and wet season, respectively. Does the "evaporation of âĹij50-80%" in P.11, Line 28 mean evapotranspiration? What is the difference between evapotranspiration and evaporation in this study?

**We revised the entire paper with regard to clearly referring to and differentiating between evaporation, transpiration, and evapotranspiration. Further, we rewrote this section and now use estimates based on the Craig-Gordon Model. The revised text reads: "The fraction of soil evaporation was estimated as 40 % from the beginning of the DS and decreased to 25 % towards the end, while it dropped from 80 to 60 % during the WS." With isotopes, we only estimated the unproductive losses from evaporation and different tools were used to estimate evapotranspiration. It is stated in the text as "We rigorously tested our results and checked their plausibility by reviewing regional data reported in the literature, and by using the CROPWAT modeling approach. Using the**

CROPWAT model (FAO 2009) forced with meteorological data for Los Baños, Philippines, we estimated an annual average reference evapotranspiration rate of 3.65 mm d–1, with a DS average of 3.96 mm d–1 and a WS average of 3.33 mm d–1."

(6) Conclusion: Seasonal distribution of soil water content and isotopic profiles was analyzed in this study, but no fluxes of unproductive soil water losses were found. Therefore, the sentence in P.13, Lines 24-25 is required to be reorganized.

**Evaporation is an unproductive water loss because apart from transpiration the rest of the water outputs from the agricultural system are considered as unproductive water losses (Bouman, 2007), and this is what we have estimated in this work.**

(7) The English writing of this manuscript should be polished further. There were some grammar errors in this paper and some sentences were confusing.

**As proposed, we will carry out an internal review again and work on the English language. A native proofreader had already checked the initial manuscript, and we were slightly puzzled when reading that the version submitted was still flawed. Nevertheless, we will send the paper again for proofreading.**

References: [1] Liu, C.M., Zhang, X.Y., Zhang, Y.Q., 2002. Determination of daily evaporation and evapotranspiration of winter wheat and maize by large-scale weighing lysimeter and micro-lysimeter. Agr. For. Meteorol. 111, 109-120. [2] Kool, D., Agam, N., Lazarovitch, N., Heitman, J. L., Sauer, T. J., Ben-Gal, A., 2014. A review of approaches for evapotranspiration partitioning. Agric. For. Meteor. 184, 56-70. [3] Zhou, S., Yu, B.F., Zhang, Y., Huang, Y.F., Wang, G.Q., 2016. Partitioning evapotranspiration based on the concept of underlying water use efficiency. Water Resour. Res. 52, 1160-1175. [4] Sprenger, M., Leistert, H., Gimbel, K., Weiler, M., 2016. Illuminating hydrological processes at the soil-vegetation-atmosphere

**References Baram, S., Ronen, Z., Kurtzman, D., Külls, C. and Dahan, O.: Desiccation-crack-induced salinization in deep clay sediment, Hydrol. Earth Syst. Sci.,**

17(4), 1533–1545, doi:https://doi.org/10.5194/hess-17-1533-2013, 2013. Bouman, B. A. M.: Water Management in Irrigated Rice: Coping with Water Scarcity, Int. Rice Res. Inst., 2007. McCutcheon, R. J., McNamara, J. P., Kohn, M. J. and Evans, S. L.: An evaluation of the ecohydrological separation hypothesis in a semiarid catchment, Hydrol. Process., 31(4), 783–799, doi:10.1002/hyp.11052, 2017. Sprenger, M., Tetzlaff, D. and Soulsby, C.: Soil water stable isotopes reveal evaporation dynamics at the soil-plant-atmosphere interface of the critical zone, Hydrol. Earth Syst. Sci., 21(7), 3839–3858, doi:10.5194/hess-21-3839-2017, 2017. Sprenger, M., Tetzlaff, D., Buttle, J., Laudon, H., Leistert, H., Mitchell, C. P. J., Snelgrove, J., Weiler, M. and Soulsby, C.: Measuring and Modeling Stable Isotopes of Mobile and Bulk Soil Water, Vadose Zone J., 17(1), doi:10.2136/vzj2017.08.0149, 2018.
* * *

---

## Author Response (AR2)

FACULTY 09

Agricultural Sciences,
Nutritional Sciences,
and Environmental Management

✉ ILR ▪ Heinrich-Buff-Ring 26 ▪ 35392 Gießen ▪ Germany

**Institute for Landscape Ecology and Resources Management**

Chair for Landscape, Water and
Biogeochemical Cycles

Amani Mahindawansha
Heinrich-Buff-Ring 26
35392 Gießen
Germany

Tel.:            0641 / 99 - 37394
Fax.:           0641 / 99 - 37389
Email: Amani.Mahindawansha@umwelt.uni-giessen.de

21.01.2020

Dear Editor,

Please find attached the revised response letters and the manuscript. You might wonder why it took so long to resubmit. First, we had communication problems among our co-authors. After we solved this, the Justus Liebig University Giessen had a severe cyber attack (no lame excuse, see https://www.uni-giessen.de/jluoffline). Since Dec 8th, we had no email system, no access to servers or the shared file systems, where our paper was stuck for weeks. We are now recovering slowly (today, January 21st, is the first day we have internet connection!). All this lead to a significant delay in the revision process of the paper. However, it also had advantages. We had sufficient time to carefully think about our initial response letters to the reviewers. Reading the response letters again, we found that there are even more things we need to change in the manuscript. Hence, we revised not only the manuscript, but also the response letters. We assume it is uncommon, but we ask that you update the response letters as well, if possible.

We are looking forward to your response. Given the limited time we had to work on the paper due to the cyber attack, we were not able to ask a professional language editing service to check the final version of the paper. But we are convinced that the revised version already improved substantially. In case necessary and in case we are allowed, we are happy to ask for a professional language check after the next round of revisions.

We sincerely hope that you will find our manuscript suitable for publication and look forward to hearing back from you.

Kind regards,
On behalf of the authors,

Amani Mahindawansha

Dear Editor,

We would like to thank you for the valuable feedback provided for our manuscript entitled, "Estimating water flux and evaporation losses using stable isotopes of soil water from irrigated agricultural crops in tropical humid regions".
Herewith we provide the answers to your comments.

Best regards,

On behalf of the authors,

Amani Mahindawansha

###

Why would the gravimetric water content be lower when using cryogenic extraction? I would expect that more water is extracted cryogenically than by oven drying.

We understand your assumption that one would expect higher values of soil water content by cryogenic extraction, particularly in our setting with 200°C extraction temperature and a high vacuum of 0.3 Pa. This is for sure the case if we take the same sample type and compare the amount of water measured via cryogenic extraction and oven drying. However, the characteristics of soil samples measured in both methods are different. The small aliquots of disturbed soil that are taken for extraction (in our case 10-15 g of mineral soil) do not have an intact pore system anymore that contains pore water. In cryogenic extraction, we therefore extract water that is immobile and attached around soil aggregates. In the case of oven drying, samples are taken via stainless steel cores (100 cm$^3$ or 250 cm$^3$). These small, but intact soil cores still have a pore system that contains pore water.

In the revised version of the paper, we will stress that we use the gravimetric soil water content from cryogenic extraction not as an absolute value, but rather as a relative value to identify differences along the soil profile.

It is still unclear where and when the ponding took place. It needs to be very clear when did you calculate evaporation from an unsaturated soil pore spaces and when the evaporation took place from open waters ponding on the fields. This would need to be accounted for when calculating evaporation fractionation as given by Gonfiantini (1986). "

All the wet rice fields are subjected to ponding. We will revise the entire paper and get rid of the misleading information on evaporation, transpiration and evapotranspiration. The revised version will focus on what we have investigated experimentally and that was evaporation. During ponding, infiltration modifies the soil water isotopic composition in the uppermost part of the profile and re-evaporation of infiltrated water has been interpreted and termed as soil evaporation. We will make the distinction between open water infiltration during ponding and soil evaporation clear.

Dear Matthias Beyer,

We would like to thank you for the valuable feedback provided for our manuscript entitled, "Estimating water flux and evaporation losses using stable isotopes of soil water from irrigated agricultural crops in tropical humid regions".  Your comments were very helpful to improve the manuscript.

Please find our point-by-point responses (in blue) to the comments (in black) below.

We believe that the modifications based on the referees' comments have resulted in an improved manuscript and hope that it is now suitable for consideration for publication as research paper in Hydrology and Earth System Sciences.

We look forward to hearing from you.

Best regards,

On behalf of the authors,

Amani Mahindawansha

In their manuscript 'Estimating water flux and evaporation losses using stable isotopes of soil water from irrigated agricultural crops in tropical humid regions' (hess-2019-213), Mahindawansha et al. investigate the effect of different crop rotations (wet rice/dry rice/maize) in seasonally flooded/ irrigated rice fields. The authors quantified the fraction of soil water evaporation in irrigated agricultural fields while also taking into account the effect of crop species and various growing stages using the Craig-Gordon model. The topic of the study is interesting and timely but, in brief, I have mixed feelings on the manuscript. While it is clearly visible that the collected dataset can be valuable for addressing the objectives of the study, there are several points that need to be addressed in order to make this contribution really valuable for the reader.

First, I have the feeling that the manuscript is lacking some internal review before publishing. The grammar is partially very poor, and I feel that several aspects (e.g. clear statement of the objectives and focus on those in the results/discussion section) should have clarified before submission. I started correcting/improving the grammar, but gave up fast on it because it became clear that major efforts are needed which I as reviewer cannot provide.

As proposed, we will carry out an internal review again and work on the English language. A native proofreader had already checked the initial manuscript, and we were slightly puzzled when reading that the version submitted was still flawed.

Second, the combined effect of Transpiration and Evaporation should be much better addressed throughout the manuscript. Recent studies proved that transpiration is generally a much greater flux compared to Evaporation, and in a study like the presented those two need to be looked at conjunctively. In that regard, also the title is confusing, because when reading 'estimation of water fluxes', one would

actually expect a water balance for the different systems, but effectively the only flux quantified is evaporation. In addition, I was confused multiple times because I was not sure if the authors speak about evaporation or evapotranspiration? (see later comments).

We agree that transpiration is larger than evaporation. In our work we look at evaporation as an unproductive loss term of the water cycle. We will revise the title accordingly (revised title: 'Investigating unproductive water losses from irrigated agricultural crops in the humid tropics through analyses of stable isotopes of water'). We will further correct the wording carefully to make the distinction of evaporation and transpiration always very clear for the reader.

Also, I was wondering multiple times if the authors really refer to soil evaporation when speaking of wet rice? If the field is flooded, it would be more open water evaporation?

Yes, of course. During ponding, infiltration modifies the soil water isotopic composition in the uppermost part of the profile and re-evaporation of infiltrated water has been interpreted and termed as soil evaporation. We will make the distinction between open water evaporation, infiltration of this ponding water and soil evaporation clear.

Having that said, I cannot recommend publishing this manuscript as is. Though the topic and study are interesting and have great potential, this is often not fully explored. With more precisely stated objectives and a subsequent focus on addressing those, I encourage the authors to improve the manuscript and increase the quality and impact of the publication. I wish the authors good luck with the revision of the manuscript.

Down below, further detailed comments can be found.

The authors state: 'None of the studies conducted so far have quantified the fraction of soil water evaporation in irrigated agricultural fields while also taking into account the effect of crop species and various growing stages. Does it make sense to calculate the evaporation from soils for wet rice, which is cultivated in a flooded system (as the authors state) → evaporation would be from open water surface anyways

Suggested objective: study the effect of crop species and various growing stages on evaporation in rotation systems

As stated above, we will fully revise the text and make sure that the terminology correct. We will also follow the suggestion and specify our objectives.

1) Title: why first singular (flux) and then plural (losses)?

We will revise the title to 'Investigating unproductive water losses from irrigated agricultural crops in the humid tropics through analyses of stable isotopes of water'

2) The abstract needs to be improved. There are many sloppy formulations and bad grammar. The results section of the abstract should be underpinned with numbers.

We will revise the abstract and add numbers to the results section.

3) What are the implications of this study and how does it help to improve management or our understanding of such systems? How to compare an irrigated/flooded rice field with a field under natural conditions in terms of water isotope interpretations?

We will add a section in the conclusion about this. It reads "

"To conclude, water losses via soil evaporation is a major unproductive loss next to leaching losses, especially during the early growing stage. Therefore, our study helps to increase understanding of soil water transport processes and evaporation losses from soil in response to crop rotation systems. Our hypothesis of reducing the unproductive water losses by introducing dry seasonal crops is supported by isotope data. Farmers should apply mitigation methods to reduce soil water evaporation, e.g. by mulching, or growing cover crops in the fallow period and by protecting the plough pan."

With regard to the second question: We compare flooded and non-flooded fields. In this sense, the dry rice and maize fields are representing the natural conditions with regard to incoming precipitation.

4) While reading the introduction, I wonder if the authors solely mean soil evaporation when they use the wording "evaporation" or if they actually mean "Evapotranspiration" (sometimes, evaporation is used for ET). The authors state that they are interested in studying soil evaporation, but can you look at one (E) without the other (T) in a combined system?

Our work is focusing on unproductive water losses as explained above. Hence, transpiration is not part of this study. To avoid further confusions, we deleted references to evapotranspiration and focus on evaporation only.

5) Methods - Extraction at 200 degrees Celsius. . .good, because very clay-rich. . .but. . .was organic contamination checked? (upper soil layers and plants)

Isotopic composition data of all water source types were checked for spectral interferences using the Spectral Contamination Identifier (LWIA-SCI) post-processing software (Los Gatos Research Inc.). None of the soil water samples were contaminated. This will be explained in the revised version of the paper.

6) Craig and Gordon modeling part should be written more concise.

We will revise section 2.4 and present the Craig-Gordon model in a more concise way.

7) What is the difference between the isotopic signal of the soil and the original isotopic signal of soil water? (do the authors mean the 'initial signal after rain/irrigation?). How justified are the assumptions made (and those are many)?

This description in the respective section was unclear. We will revise this section in the following way: "In our study, the original isotopic signal $\delta_p$ is the signal of the water input via precipitation or irrigation. During the WS, $\delta_p$ was estimated as the weighted average of the isotopic signals from the most frequent large precipitation events. For the DS, we used the weighted mean of the irrigation water as the input signal."

8) For the results, it would be interesting to see if the fraction calculation fits with the modelled results

We revised the entire section and deleted the results of the modeling approach as they were not directly comparable to the experimental results we obtained. The reason is, that the CROPWAT model does not provide information on evaporation, but only on evapotranspiration.

9) P1.l. 13 advance better: improve

Will be corrected.

10) P1.l.18: progressed through the growth – bad grammar

Will be corrected.

11) P1.l.23 compared to over

Will be corrected.

12) P2 l.6-11: not only in recent years, this has been studied since Allison et al. in the 80's. . .has not been studied as much compared to what?

We will revise this section.

13) p.2.l.13 it p.2.l.20-32: this is well-written!

Thank you!!!

14) p.3.l.5: Our objectives during this study are the objectives of this study are

Will be corrected.

15) p.3. l. 5-8: Objectives should be formulated clear and concise

We will revise the objectives.

16) p.3. l. 21: constancy consistency

Will be corrected.

17) p.3. l. 20-23: if the mung bean plot was not used it is not necessary to mention it here

This sentence has been deleted. We will include a note in the figure caption of Fig. 1 why mung bean plots are depicted in the figure.

18) p.4. l. 6: the model controls?...grammar

Will be corrected.

19) p.4. l. 8: mixing from macropore flow from cracks?...grammar

Will be corrected.

20) p.6. l. 25/26: the shape of the isotopic profiles in the shallow soil water changed depending on the crop and growth stage. → only because of that or also other factors – irrigation water isotope values, precipitation, radiation?

We agree that this statement was incomplete and hence, we will delete it. In the revised version of the manuscript, we will streamline the paper according to the main objectives.

21) What are the conclusions of the authors regarding the magnitudes of evaporation? (Are these numbers given as fraction of total evapotranspiration?)

Yes, the fractions of evaporation are given as percentages of total evapotranspiration. The magnitude of evaporation is higher at the beginning of the growing period and decreases towards the end of the season.

We revised the section accordingly and explain this in more detail and discuss our results in relation to other published values.

22) Fig. 3: bad resolution

We submitted high-resolution graphs separately in the first submission, but obviously, they have not been included in the PDF. We hope it will work out this time.

23) Section 4.3.: the statements here are very interesting and it is appreciable that the authors introduce this discussion. unfortunately they question parts of the isotopic data presented in the study

We think that potential limitations of our method should be discussed. It also directs for future researches. Nevertheless, the effect discussed does not question the study in general as it affects only a limited set of samples. We will make this clear in the revised manuscript.

24) Conclusion: - throughout the manuscript, the phrase 'redistribution via plants/roots/etc. appears frequently', but it is not discussed anywhere. I suggest leaving this out or providing further evidence. - 'the conclusion that isotopic profiles develop via diffusion processes in the shallow soil and are then transported by advection in the matrix or in macropores or cracks' → please rephrase, poor grammar

We agree with the reviwer and we will include a new section on this process in the discussion. It will read: Further, hydraulic redistribution of water in the vadose zone is an important process of passive transport of soil water along a hydraulic gradient through the rooting system (Richards and Caldwell, 1987). Therefore, hydraulic redistribution can influence the pore water stable isotopic composition and reshape the soil water isotopic profile. Sprenger et al. (2016) discussed the significance of hydraulic redistribution in the soil hydrological cycle. However, the influence of hydraulic redistribution on the isotopic composition is likely very small (Walter, 2010).

Apart from that, we will revise the grammar in the conclusions.

25) p.13., l. 13: Do the authors really mean Evapotranspiration or rather Transpiration?

We meant transpiration fraction of the evapotranspiration and we will correct it in the revised version.

Dear reviewer,

We would like to thank you for the valuable feedback provided for our manuscript entitled, "Estimating water flux and evaporation losses using stable isotopes of soil water from irrigated agricultural crops in tropical humid regions". Your comments were very helpful to improve the manuscript.

Please find our point-by-point responses (in blue) to the comments (in black) below.

We believe that the modifications based on the referees' comments have resulted in an improved manuscript and hope that it is now suitable for consideration for publication as a research paper in Hydrology and Earth System Sciences.

We look forward to hearing from you.

Best regards,

On behalf of the authors,

Amani Mahindawansha

Review of Hydrology and Earth System Sciences Manuscript: hess-2019-213 Title: Estimating water flux and evaporation losses using stable isotopes of soil water from irrigated agricultural crops in tropical humid regions Authors: Amani Mahindawansha et al.

This manuscript presents seasonal variations in the soil water isotopic profiles and the fraction of evaporation (FE) for different crops (wet rice, dry rice and maize) under flooded and non-flooded irrigation management practices. This topic is interesting for understanding water cycle and water conservation in agricultural fields. However, there are some issues within the manuscript that requires substantial interpretation and improvement. The following is my detailed comments.

 (1)  Abstract:

Since only FE values were calculated and no water flux of evaporation were determined in this study, the second sentence (P.1,Lines 13-15) should be changed.

We agree and will revise not only the abstract, but also the title.

Other evidences should be given to prove the occurrence of piston type matrix flow or preferential flow besides the isotopic data in the text (P.1, Lines 22-24).

We will revise the abstract. The discussion of the different flow mechanisms will be more general.

It is helpful to supplement important data in the abstract section to clarify the new findings of this study.

We agree with the reviewer and added the main results to the abstract.

(2) Introduction:

Determination of the soil evaporation flux (E) and the fraction of E in ET (FE) have been widely studied using several methods and techniques for different irrigated crops (Liu et al., 2002; Kool et al., 2014; Sprenger et al., 2016; Zhou et al., 2016). The new scientific merits in this study are not very clearly clarified.

The merit of this study is quantifying the fraction of soil water evaporation in irrigated agricultural fields and taking into account the effect of different crop species and different irrigation and management practice at various growing stages. The references listed will be considered in the revised version of the introduction. To better account for previous work, we will include the following sentences: "
The determination of soil evaporation and the fraction of evaporation in relation to total evapotranspiration have been widely studied using several methods for different crops. For example, Liu et al. (2002) studied evapotranspiration from winter wheat and maize, using weighing lysimeters. Zhou et al. (2016) partitioned evaporation and transpiration fluxes for corn, soya bean, grassland and forests using flux tower measurements. Kool et al. (2014) applied different methods such as chamber, micro-lysimeter, and soil heat pulse to estimate the evaporation and used stable isotopes of water to separate evaporation from transpiration.."

We do not agree that the research gaps of current studies were not clearly stated. We would like to draw the attention to the end of the introduction where we stated: "None of the studies conducted so far have quantified the fraction of evaporation losses in rice-based cropping systems, taking into account the effect of crop species and various growing stages."

(3) Material and Methods:

There are straw and non straw applications conducted for different treatments in the experiments (P.3,Lines 20-21). How does the straw application affect the seasonal variations in the FE for different irrigated crops?

We did not find significant differences between the isotopic compositions of soil water with or without straw application. Therefore, we pooled the results for each crop. See section 2.4 "The isotopic values of the two treatments straw and no-straw application as a control plot were combined for each crop for further analysis, as there were no significant differences for stable isotopes of water between the treatments (p>0.05)."

One would have expected a stronger effect of straw application on evaporation. However, the straw was not applied as a typical mulch layer to reduce evaporation, but was partly worked into the soil to reduce crack formation. This information is now included in section 2.2.

Please describe in detail how to determine the time when a water shortage occurred in dry rice and maize fields (P.3,Lines 26-27).

Field workers from the IRRI were responsible for watering dry crops in times of soil water shortage. The decision of watering was not set by specific thresholds or indicators, but by expert knowledge. This information will be added to the text.

The gravimetric soil water content is determined traditionally by oven-drying method. Smaller values might be resulted by using the soil water loss in cryogenic water extraction process to determine the soil water content (P.4,Lines 19-20).

We agree that the results of both methods are not directly comparable. We added a sentence to section 2.2 to address this: "Soil water content determined this way deviates from the classical oven drying method and results in slightly lower values. In the case of oven drying, samples are taken via stainless steel cores. These soil cores still have an intact pore system that contains pore water. Pore water is not captured by cryogenic extraction. However, we use the gravimetric soil water content from cryogenic extraction not as an absolute value, but rather as a relative value to identify differences along the soil profile."Root length density was analyzed as described in the P.4, Lines 21-22 in the "Material and Methods" section, but non detailed results were shown in the "3 Results" section.

The description of the method to measure root length density was a remnant from a previous draft of the manuscript which we forgot to delete. We will remove it in the revised manuscript.

 (4)  Results:

The lc-excess was developed/introduced by Landwehr and Coplen (2006) in respect to River Water Line. They used the lc-excess to determine how the isotopic values of river waters differed from their sources (i.e., precipitation). However, the authors use lc-excess to estimate the deviation in the isotopic values of the soil samples from regional precipitation. I do not find any good argument why the authors use lc-excess since there is no river water sampled during their experiments. The lc-excess is not necessarily needed in this study (P.7, Lines 9-16). Instead, the deviation of soil isotopic values from LMWL/GMWL is already indicating the evaporation process and it is more commonly used method. Lower indicates condensation process and higher indicates evaporation process.

Sprenger et al. (2017) stated "…that lc-excess was advantageous over the deuterium-excess (or single isotope approaches with δ2H or δ18O) for inferring evaporation fractionation, because the lc-excess of the precipitation input is about 0‰ and with relatively little seasonal dynamics, while δ2H, δ18O, and d-excess can have an intense seasonal variability.". As we analyzed soil profiles from different seasons, seasonality effects might be a problem. To avoid this, we used the lc-excess for comparison. Furthermore lc-excess has been successfully applied it for soil water studies, previously by Sprenger et al. (Sprenger et al., 2016, 2017, 2018), McCutcheon et al. (2017) .

(5) Discussion:

The authors estimate the annual average reference evapotranspiration rates in dry season and wet season, respectively. Does the "evaporation of ∼50-80%" in P.11, Line 28 mean evapotranspiration? What is the difference between evapotranspiration and evaporation in this study?

We deleted the section about the CROPWAT model as the model and our field experimental work are not direct comparable. This section was misleading. Furthermore, we only address the evaporation process.

(6) Conclusion:

Seasonal distribution of soil water content and isotopic profiles was analyzed in this study, but no fluxes of unproductive soil water losses were found. Therefore, the sentence in P.13, Lines 24-25 is required to be reorganized.

Evaporation is an unproductive water loss because apart from transpiration the rest of the water outputs from the agricultural system are considered as unproductive water losses, and this is what we have estimated in this work. That will be added to the introduction as follows:

"Unproductive water losses are those that do not lead directly to biomass production, such as transpiration, and include for example leaching, evaporation from the soil or from ponding water (Bouman, 2007)."

(7) The English writing of this manuscript should be polished further. There were some grammar errors in this paper and some sentences were confusing.

As proposed, we will carry out an internal review again and work on the English language. A native proofreader had already checked the initial manuscript, and we were slightly puzzled when reading that the version submitted was still flawed. Nevertheless, we will send the paper again for proofreading.

[revised manuscript text omitted]

---

## Author Response (AR3)

✉ ILR ▪ Heinrich-Buff-Ring 26 ▪ 35392 Gießen ▪ Germany

**Institute for Landscape Ecology and Resources Management**

Chair for Landscape, Water and
Biogeochemical Cycles

Amani Mahindawansha
Heinrich-Buff-Ring 26
35392 Gießen
Germany

Tel.:            0641 / 99 - 37394
Fax.:           0641 / 99 - 37389
Email: Amani.Mahindawansha@umwelt.uni-giessen.de

12.05.2020

Dear Editor,

Please find attached the revised response letters and the manuscript. We would like to thank you and the reviewer for the valuable feedback provided for our manuscript entitled "Investigating unproductive water losses from irrigated agricultural crops in the humid tropics through analyses of stable isotopes of water".

We believe that the modifications based on the reviewer and your comments resulted in an improved manuscript. We sent out the manuscript for professional proofreading. We sincerely hope that the manuscript is now suitable for consideration for publication as a research paper in Hydrology and Earth System Sciences and look forward to hearing back from you.

Kind regards,
On behalf of the authors,

Amani Mahindawansha

Dear editor,

We would like to thank you for the valuable feedback provided for our manuscript entitled, "Investigating unproductive water losses from irrigated agricultural crops in the humid tropics through analyses of stable isotopes of water". Your comments are very helpful to improve the manuscript. Please find our point-by-point responses (in blue) to the comments (in black) below.

We believe that the modifications based on the referee and editor comments result in an improved manuscript. We also sent out the manuscript for professional proofreading. We hope that the manuscript is now suitable for consideration for publication as a research paper in Hydrology and Earth System Sciences.

We look forward to hearing from you.

Best regards,

On behalf of the authors,

Amani Mahindawansha

#####################

Editor Decision: Reconsider after major revisions (further review by editor and referees) (26 Mar 2020) by Matthias Sprenger

Comments to the Author:

I thank Amani Mahindawansha and colleagues for their revisions under the difficult conditions of the situation at the University of Giessen.

I received two reviews of the revised manuscript of which one reviewer accepted the manuscript as is and another reviewer provided an extended list of feedback with a request for major revision.

I also thoroughly read the revised manuscript and have several aspects that need to be addressed before publication.

Please respond in your revision to each comment and provide examples of how the manuscript has been changed to address the feedback.

##########

I do not think that the spatial information as shown in Figure 1 of the experimental set up is that important. One assumes that the proximity between different treatments does not affect the results. However, the temporal dynamics of the treatment are still not clear to me. For example, when was the field flooded? For how long were they flooded? Which samples were taken during flooding (you state in P4L10 that some samples were taken during flooding conditions). When did you assume n=1 and when n= 0.5 in the C-G model?

Figure 1 was added upon request during previous round of reviews.

Wet rice fields were flooded during both seasons. Therefore, all the wet rice samples were taken during flooding. We have mentioned that in the text as: "*Wet rice fields were maintained at water-flooded conditions, except for the first and last two weeks between transplanting and harvest (Fig. 2).*" Temporal information on water management, transplanting, seeding, harvesting, as well as sampling is depicted in figure 2. We further improved this figure and included information on the duration of

the flooding indicated by a blue horizontal bar. We now refer to this information also in the text, of the revised manuscript.

Further information on the settings for the aerodynamic diffusion parameter *n* is now provided on P6L7-10: *"As part of the calculation of εk, the aerodynamic diffusion parameter n [–] has to be set. It reaches 1 when the soil is dried to residual moisture levels (Mathieu and Bariac, 1996), presenting turbulent conditions. We anticipated n=0.5 for wet rice fields with saturated soils (Good et al., 2014), n=0.7 for dry rice, and n=0.9 for maize."*

P2L14: General enrichment of soil water in heavy isotopes could also be a result of infiltration of rainwater that. The deviation of the soil water from the LMWL is an indicator (= kinetic fractionation). Yes, equilibrium fractionation might take place within the soil, were humidity is high,. Please rephrase.

Yes, but here we particularly refer to dry periods where the rainfall frequency and amounts are very low. Mixing with other waters is referred to in the previous sentence. We make this clearer in the revised version, *"During dry periods, the isotopic enrichment of shallow soil water is generally driven by evaporation (Gangi et al., 2015; Liu et al., 2015) and is affected by equilibrium and kinetic fractionation (Gat, 1996; Gonfiantini, 1986). "*

P2L16: Benettin is not the correct reference here. Please refer to older manuscripts (e.g. by Gat, Gonfiantini).

Corrected as recommended.

P3L6: Consider rephrasing to emphasize the focus on agricultural soils. Currently, for (II), it is held very general, but I suggest to rephrase (II) and (III) to include the focus on agricultural soils in both objectives.

We added *"…of agricultural soils"* to objective II. Objective III already indicated agricultural fields.

P3L12: If these precipitation amounts are reflecting the specific study years, this needs to be stated. If they are long-term averages, this also would need to be clarified.

The data cover the study period 2015 and 2016. We revised the text accordingly.

P3L14: Please put the study periods in perspective. Where they exceptionally dry/wet or representing long-term climate is enriched in heavy isotopes observations?

We included the following sentence to provide further information: *"Both seasons represented typical weather conditions in the region."*

P4L18: What does "Pore water is not captured by cryogenic extraction" mean?

The sentence was revised and now reads *"However, as soil samples taken for cryogenic extraction are disturbed soil samples, they do not include all the pore water."*

P4L20: How was GW sampled? At which depth is the water table?

We included now further information on the groundwater sampling and condition: *"Groundwater and ponded surface water of flooded rice were collected once a week from each plot at existing sampling stations (Heinz et al., 2013). Low-cost car wiper pumps (Art. Nr. 103 158, TOPRAN, Bremen, Germany) were used to pump water with a pumping rate of 40 L h-1 to the sampling container which was installed in the centre of the nine fields (Fig. 1). The installation length below ground was 2.0 m. The groundwater table varied depending on the season. During the WS the groundwater table was at 0.5-0.6 m below ground and at 0.6-1.7 m during the DS. Rainwater and irrigation water were sampled event-based. For*

*detailed information on the experimental design and sample collection as well as field preparation such as puddling and ploughing, see Mahindawansha et al. (2018b, 2018a) "*

P4L24: were

We could not find any error.

P5L7: Unclear why "This isotopic signal is then carried to deeper compartments via leaching. In deeper compartments, mixing of soil water with water transported through cracks may occur. The concept of multi-compartment transport indicates the history of the evaporation process as well as the depth and degree of isotope signal changes by the preferential flow." is mentioned here.

We agree that this section does not fit here and have therefore edited and moved it to the discussion.

P5L16: Most frequent? What does this mean? Do you mean the last event before sampling?

During the WS, there were many large rain events and few very small rain events. Here most frequent means the frequently occurred larger amounts. We edited the text as *"...frequently occurring large precipitation events (larger than 10 mm)."*

P5L19: (2018) twice

Corrected.

P6L1: Again Benetting is not the correct reference here. I believe that n=1 is giving for example in Gonfiantini or in Allison. Also, this indicates that for wet rice, there was always ponding. Please clarify when which n-value was used.

See comment above. Text has been revised (P6L7-10).

P6L2: How sensitive are the calculations to changes in the assumed n-value?

The standard deviations of the fractions due to n=1 and n=0 are 0.01 for 2H, and 0.05 for 18O, and therefore the sensitivity can be considered as very low/negligible.

P6L13: I suggest to keep the expression of "surface ponded water", as introduced in your methods section to prevent confusion in the results section, because "surface water" could as well be stream water for example.

We followed the suggestion.

P7L12: What does "pattern .... Increased" mean?

We revised this section to make our statements more clear. The text now reads: *"We found an exponential increase in the lc-excess along the soil profile, particularly for maize, but also, though less apparent, for dry rice soils (Fig. 3l, p). For wet rice during the DS, the exponential pattern was even less obvious, but shallow soil layers still depicted lower lc-excess between -10 to -5 ‰ than deeper soil layers with values of -5 to 0 ‰. In contrast, lc-excess values of shallow soils in wet rice fields of the WS (Fig. 3d) generally decreased with depths of up to 20 cm and then levelled out at around -7 to -9 ‰. These patterns indicate a higher evaporation signal in shallow soils for the DS crops compared to the WS crop. The highest evaporative fractionation was found near the surface in maize fields with significantly lower lc-excess values during the last growing stage GS3..."*

P7L13: Please add GW, irrigation and rainfall to the lc-excess plots.

Added as recommended.

P7L14: This sentence is an interpretation and not presenting results. Consider moving to discussion section.

We reformulated and moved the sentence to the beginning of the discussion.

P7L19: Precipitation and irrigation isotope ratios should be presented at the beginning of section 3.1

We moved this information as recommended.

P7L20: This sentence seem to be not correct. "Frequent precipitation events" could also have relatively similar isotope ratios. Therefore, this sentence need to be rephrased.

We rephrased the section, see our reply to the comment on P7L12.

P7L33: Unclear what ", and along with depth towards deep soils (0.20±0.1)" means.

Thanks for indicating this. Reading the section again, we must admit that it was difficult to capture. Hence, we revised the entire section (3.2). It now reads: *"The estimated fraction of evaporation FE at each soil depth was derived by means of an evaporative enrichment of heavier isotopes in the soil water. Fig. 5 shows FE estimated based on both isotopes for the growing seasons WS and DS, the growing stages GS1-GS3, the different crops wet rice, dry rice, and maize as well as for different soil depths. A clear trend of FE with soil depth can be depicted at all growing stages during the DS for both crops, maize, and dry rice, reaching below 0.2 for dry rice and even below 0.1 for maize in deep soils (Fig. 5g-l). During the DS, soils in dry rice fields showed high soil FE at shallow depths at the beginning of the first two growing stages GS1 (0.54±0.1) and GS2 (0.50±0.1) which decreased to FE of around 0.27±0.1 at GS3. Further, we observed lower FE average values (0.2±0.1) in deep soils between 0.25 and 0.6 m in these fields. For maize, FE remained stable at 0.3±0.1 in shallow soils throughout the season and decreased with depth for both isotopes (to around 0.07±0.05) (Fig. 5j, k, l). The FE in shallow soils of wet rice in the DS ranged from 0.42±0.08 to 0.20±0.08 (similar for both isotopes) and remained nearly stable in deep soils at 0.13±0.1 (Fig. 5d, e, f). Overall, we did not find a similar decreasing trend with depth, as reported for dry rice and maize. Instead, particularly during GS2 and GS3, the highest fractions of FE were found at moderate soil depths of 0.1 to 0.2 m. Results regarding the estimation of FE based on δ2H and δ18O are fairly similar for all dry season crops. For wet rice in the WS, FE of both isotopes differed significantly by 0.1 to 0.2 for the top soil and even 0.5 for deep soils (Fig. 5a, b, c). During the WS, FE in shallow soil decreased from around 0.72±0.1 (GS1) to 0.47±0.06 (GS3) for δ2H and from 0.87±0.07 (GS1) to 0.76±0.07 (GS3) for δ18O. The general trend with slightly higher FE at moderate soil depths and again decreasing FE further down, which we observed for wet rice during the dry season, was also confirmed for wet rice during the wet season. The soil water in wet rice fields during the WS carried a larger signal of high evaporation losses down along the soil profile. The estimated FE from ponding surface water (data not shown in Fig. 5) was found to be larger during the WS than during the DS with no significant difference between δ2H and δ18O. The FE of ponded water during the WS did not fluctuate with time, and remained close to 0.92±0.07, while during the DS values decreased from GS1 (0.67±0.03) to GS3 (0.24±0.01). Here, FE of ponded surface water indicates a high evaporation loss during the WS. The evaporation signal is carried to deeper layers by subsequent infiltration and percolation. "*

P7L21: varied

Corrected. Thanks

P8L17/18: "separation"? You mean fractionation?

No, we meant the change of the isotopic pattern, separating the soil into shallow and deep soil. However, we revised this and the sentences before. The section now starts as: *"Depending on the evaporation effect on soil water isotopic composition and water transport processes, we found a change of the isotopic composition at around 0.2 m below the surface at our study site. "*

P8L20: Ploughing and puddling practices should have been introduced in the site description.

We have now introduced the terms in the site description, but do not further explain any detailed field preparation as this is not in the focus of this paper. However, we have guided the reader to check the paper Mahindawansha et al. 2018b for more details about site description including land preparation.

P8L26: It is not clear how they are transported downwards. You are mentioning the drying stage, but water movement during drying is minimal, since pressure differences in the soil will have been already equilibrated by then. Or do you mean the downward movement of the drying front?

We have mentioned here that how the isotopic profile is affected by heavy water (enriched) which transported downwards in dry rice and maize field. We revised the text to make it clear. *"In the unsaturated zone in dry rice and maize fields, the diffusive vapour transport process is dominant (Bittelli et al., 2008). Kinetic fractionation leads to the accumulation of heavy water molecules (formed by $^2H$ and $^{18}O$) at the water-air interface, which are subsequently transported downwards and then mixed with the soil matrix (Horita et al., 2008). Downward water movement at steady-state or slowly changing conditions results in an exponential evaporation profile with depth during the drying stage that is comparable to those found in soils beneath dry rice and maize (Fig. 3i, j, m, n) (Zimmermann et al., 1966; Barnes and Allison, 1988; Rothfuss et al., 2015). "*

P9L8: It would be very helpful to see the lc-excess of the ponding water and irrigation water in Figure 3 d,h,l,p to support the "single compartment" hypothesis. Also, in Figure 3d, the lc-excess does not support a single well-mixed compartment, since the lc-excess varies within the upper 20 cm.

We included the missing information in Fig. 3. Further, we deleted the sentence regarding the single compartment, as it was misleading.

P9L8: How do you derive the infiltration front. What is the depth of it that you are referring to here?

We have revised this section, see comment above.

P9L11: If the conclusion is that "piston-like matrix flow" is dominant, what did you mean with "single compartment"? I guess this expression is unclear.

Again, see comment above.

P9L15: The groundwater depth information should already be mentioned in the site description (i.e., it is important to know that the soil water sampling generally took place above the GW table.

We added the information.

P9L15: If capillary rise takes place, why is the bottom of your isotope depth profile so different from the groundwater isotope ratio? You refer to isotopically depleted GW, but your observations of GW are enriched compared to the soil.

We understand that our explanation was unclear. We therefore changed the text. It now reads: *"In soils with fine pores, capillary rise could have further affected the observed isotopic patterns, depending also on depth to groundwater. It has been shown that capillary rise of shallow groundwater can influence soil moisture and its isotopic composition in the upper meter of clayey soil (Baram et al., 2013; Clark and Fritz, 1997). An upward matrix flow through capillary rise has probably occurred in our system as well, given the fine texture of the soils (Table 1). However, the effect seems to be negligible, as the*

*GW signatures we measured were more enriched than the soil water found at greater soil depths (Fig. 3). This is probably due to the fact that GW head levels were often substantially below the deepest soil layer we sample (Mahindawansha et al., 2018a)..*"

P9L16: Not sure why hydraulic redistribution would result in a smoothing of the isotope depth profiles. If deep roots would transport water to shallow depths, one would expect a spike of deep water isotope ratio in the otherwise shallow water isotope ratio.

Smoothing was a misleading term. We revised the section to improve our explanation as follows: *"The observed isotopic signals in the shallow soils could also indirectly be explained by a transpiration bias. Transpiration decreases the soil moisture, but preserves the isotopic composition (Baram et al., 2013). With decreasing soil moisture, incoming water has a relatively stronger imprint on the soil's isotopic composition. "*

P9L31: Unclear why the irrigation water has an "evaporation imprint".

We revised the section for better explanation. We also provided further information on the source of irrigation water in the Material and Methods section: *"Note that irrigation water was taken from an open reservoir, located next to the fields. The reservoir is regularly filled with groundwater that is characterized by a uniform seasonal composition with an isotopically depleted characteristic (Mahindawansha et al., 2018a). "*

P10L2: It is unclear why preferential flow would be a necessary process to reach to "gradual isotopic depletion towards deep soils". Based on simulations assuming no preferential flow (see Fig. 9 in Sprenger et al., 2019, doi: 10.1002/2015RG000515), the mixing of fractionated soil water with non-fractionated rainwater would result in similar profiles as you show. Unclear how you can derive preferential flow from the observations.

Based on field observations we know that irrigation water reaches deeper soils via preferential flows (Mahindawansha et al., 2018), and then mixes with soil water. He et al. (2017) have also observed leaching losses of water and nutrients in a lysimeter experiment, which they attributed to crack flow mechanisms in the same study site. Although the shape of the profiles is similar to that reported by Sprenger et al. (2019), our current process understanding is reflected by an explanation considering preferential crack flow. Therefore, we would like to stick to our current description.

P10L10: Here you explain the shape of the isotope depth profile with evaporation, while before you explain the shape with preferential flow.

We have categorized the isotopic soil water patterns we observed mainly into two parts, those found at shallow (0 – 0.2 m) and deep soils (0.2 – 0.6 m). The overall shape of soil water isotope patterns is a combination of all the processes going on in both parts, which are hard to distinguish. However, preferential flows affect soils mainly below 0.2 m as water moves faster along the cracks and rapidly reaches deeper soils (in maize fields). Contrary, Shallow soils are mainly characterized by evaporation processes. We have revised part of the sections (4.2 and 4.3) to make clear that the evaporation effect is mainly responsible for the isotopic pattern observed between 0-20 cm soil depth.

P10L17: I suggest to not cite (Sprenger et al., 2016), but a classic paper from the first generation isotope hydrologists describing such basic concepts (e.g., Dansgaard).

Edited as recommended.

P10L26: Unclear what is meant with "profiles with multiple compartments"

Changed to "*multiple soil layers*".

P11L13: leaf area

Corrected.

P11L16: Rephrase to clarify that you are talking about your own data and not Rothfuss anymore.

We rephrased the sentence as "*In our study, the fraction*..."

P11L18: reported for Asia? What kind of climate, soil and vegetation? 30% of F_E for Asia is not a very meaningful information.

We agree that the previous description was too vague and included more references. The revised text now reads: *"Values of about 30 % (maize) and 50 % (dry rice) evaporation losses were reported for Asia at the same sites in the Philippines based on eddy covariance measurements by Alberto et al. (2014). Similar values were confirmed by Bouman et al. (2005) based on a review of tropical upland and lowland rice varieties under irrigated aerobic conditions. Simpson et al. (1992) report 40 % for flooded rice fields in a semiarid region of south eastern Australia and Maruyama and Kuwagata (2010) about 60% for paddy fields in southwestern Japan.* "

P12L21: You mean aerobic conditions?

No, it is anaerobic. We added that to the sentence.

P12L27: errors related to F_E?

As Rothfuss et al. 2010 explains, maximum uncertainty of the partitioning is much larger and ranges from 1% to 29%, depending on the value of $\alpha_k$ and the day of the partitioning. We added this information to the text for a better understanding as: *"Determination of $\alpha_k$ can also result in estimation errors (i.e., a maximum uncertainty of the partitioning) of 1 to 29 %, depending on the value of $\alpha_k$ and the day of the partitioning (Rothfuss et al., 2010).* "

P12L30: For now, I am not convinced that your data provides sufficient info about preferential flow. Also, (III) is part of (II): preferential flow is part of "soil water movement".

We have revised the manuscript in accordance with the comments made before. Several of these comments dealt with preferential flow. We hope that we have made it clear that preferential flow through cracks is an important water flux pathway in these rice-based cropping systems, which is backed by data we provide here and previous work published (Mahindawansha et al. 2018a,b).

We see your other point and therefore we rephrased the text better separating the three major water flux processes. We have further deleted the Roman numbers to avoid confusion with the Roman numerals of the objectives in the Introduction.

P13L2: What are "lower compartments"?

Changed to "*deep soil layers*".

P13L4: How are these two sentences related? Why using "However"?

We deleted "*However*".

P13L7: Something seems to be missing here: not able to measure?

Sentences is revised.

Figure 2: What is the x-axis scale? Please add the measurement frequency mm/h or mm/day? Is it correct that on a day that wet rice experienced irrigation there was no irrigation for dry rice or mize? Some seem to overlap, but

Measurement frequency is added (mm/day). Yes, it is correct, wet rice is more often irrigated than dry rice and maize.

Figure 3: To my understanding, you measured gravimetric water content. Please use that instead of soil water content, which is usually used for volumetric water content. I also suggest to use the unit g g-1 in this context.

We changed the wording in the text and adapted Figure 3 and its caption accordingly.

Figure 4: Description of the regression line is missing in the caption and/or legend

We added this information into the figure caption.

Dear reviewer,

We would like to thank you for the valuable feedback provided for our manuscript entitled, "Investigating unproductive water losses from irrigated agricultural crops in the humid tropics through analyses of stable isotopes of water". Your comments are very helpful to improve the manuscript. Please find our point-by-point responses (in blue) to the comments (in black) below.

We believe that the modifications based on the referee and editor comments result in an improved manuscript. We also sent out the manuscript for professional proofreading. We hope that the manuscript is now suitable for consideration for publication as a research paper in Hydrology and Earth System Sciences.

Best regards,

On behalf of the authors,

Amani Mahindawansha

##############

In this paper, Mahindawansha et al. uses the variation of soil isotopic signature between depth, seasons and crop management practices (dry rice, wet rice, maize) to infer mixing processes and fraction of evaporative losses.

This is an interesting topic with significant implications for sustainable management practices in agrosystems, and the study is based on an extensive and valuable datasets.

While the underlying study has good potential for publication in HESS, I think the manuscript needs substantial improvement in the way the study is presented, and the results analysed.

Above all, the level of English needs to be improved (as already noted in the first round of review), also because too many sentences can lead to confusion for the reader.

Below are some specific comments.

Specific comments

P1L16-17: To me it seems that wet rice (WS) does not fit in this description (Fig. 3a-b): the shallow larger are depleted with respect to deeper ones

We agree that our description was too short. We revised it to. "*For dry rice and maize, water in shallow soil layers (0 to 0.2 m) was more isotopically enriched than in deeper soil layers (below 0.2 m). This effect was less pronounced for wet rice, but still evident for the average values at both soil depths and seasons.*"

P5L11: The fraction of evaporation losses to...what? Total evapotranspiration? Accumulated infiltration? Please provide a more detailed description, as this is one the key variable discussed in this study.

The fraction of evaporation is related to the total amount of soil water. We have revised the sentence, it reads: "*Equation 1 is based on the Craig–Gordon model and formulations introduced by Gonfiantini*

*(1986) to estimate the fraction of evaporation loss (FE) from the soil water based on an isotope mass balance approach as follows."*

P5L23-24: If the authors assume equilibrium, can they explicitly provide the relationship for the reader? I am assuming it is \delta_A = (\delta_{rain}-\epsilon^+)/\alpha_+.

Yes, correct. The equation is mentioned in the supplementary material, which is now cited in the respective lines of the manuscript.

P6L7-8: I do not understand this sentence.

We have shortened and clarified the description of the statistical analyses. It now reads: "*We tested for significant statistical differences (p≤0.05) of stable isotopes of water (δ2H and δ18O) during seasons, growing stages, and treatments between all water sources. Normal distribution was tested by the Shapiro Wilk test and homogeneity of variances by the Fligner Killeen test (Python 2.7.10.0). Because of the non-normal distribution of data, we further carried out a non-parametric rank-based test considering no ties. The isotopic values of the two treatments with straw and without straw as a control plot were combined for each crop for further analysis, as there were no significant differences for stable isotopes of water between the two treatments (p>0.05).*"

P6L13-14: Do you show this somewhere in the tables / figures?

Yes, it is shown in Fig. 3. We added that to the text.

P6L4-15: A figure with boxplots, instead of Table 2, would be more direct for interpretation.

These results are already plotted in Figure 3. Due to the detailed nature of Figure 3 and the amount of information included there, we decided to additionally provide the information about the different water samples in a table.

P6L22-24: Can you be more precise? Stating for example that overall the soil tend to be more depleted in ^2H and ^{18}O in WS than in the different WS cases. The same remark applies to the second sentence about maize, wet rice and dry rice.

We added further information on results related to Figure 3 in the revised manuscript as suggested. The section reads: "*The isotopic composition of soil water from fields with different crops during the DS were statistically different (more enriched) from the wet rice during the WS. Within the DS itself, there was a tendency for more depleted conditions in the upper soil horizons of wet rice compared to maize and dry rice. We did not find such a distinct difference for the soil layers below 0.2 m. Results for GS2 and GS3 of maize and wet rice were statistically different during the DS, and maize and dry rice were statistically different except for the GS3 of dry rice.* "

P7L8: I suggest using a less ambiguous formulation: "lc-excess is an indicator of evaporative fractionation, with more negative values here reflecting larger losses from soil evaporation." Note that this sentence could be moved to the end of section 2.3, where lc-excess is defined.

We moved the sentence as recommended.

P7L8-10: To me it seems that wet rice (WS) does not fit in this description: for each GS stage, lc-excess decreases with depth, and then stabilized below ~20cm.

Wet rice (WS) has not been mentioned here, but is referred to at the end of that section. Nevertheless, we have revised this entire section. It now reads: "*We found an exponential increase in the lc-excess along the soil profile, particularly for maize, but also, though less apparent, for dry rice soils (Fig. 3l, p). For wet rice during the DS, the exponential pattern was even less obvious, but shallow soil layers still*

*depicted lower lc-excess between -10 to -5 ‰ than deeper soil layers with values of -5 to 0 ‰. In contrast, lc-excess values of shallow soils in wet rice fields of the WS (Fig. 3d) generally decreased with depths of up to 20 cm and then levelled out at around -7 to -9 ‰. These patterns indicate a higher evaporation signal in shallow soils for the DS crops compared to the WS crop. The highest evaporative fractionation was found near the surface in maize fields with significantly lower lc-excess values during the last growing stage GS3. For maize, the lc-excess values decreased in most soil layers from GS1 to GS3, which was the opposite for dry and wet rice during the DS. No distinct, clear pattern could be found along the growing stages for wet rice during the WS."*

P7L10-11: "the highest evaporation" ? I do not think the authors can compare evaporative fluxes between cases solely based on lc-excess values. It would be more accurate to say "the highest evaporative fractionation".

We edited as recommended.

P7L14-15: The sentence seems a bit confusing or redundant, can the authors reformulate? Alternatively, it could be removed, as it does not add much to the description of Fig. 4 below.

Removed as suggested.

P7L24-25: This is an interesting interpretation, but without data from the past DS, it is quite speculative. I suggest removing this sentence, leaving it for the Discussion (as is mentioned in P10).

Removed as recommended.

P8L4-5: Do the authors mean a "significant" difference between $\delta^2H$- and $\delta^{18}$-derived $F_E$, or between growing stages? I am guessing it is the former, but there also a large seasonal difference between GS1 and GS2 (the latter similar to GS3). Please clarify.

Here we meant the seasonal differences between WS and DS. The entire section 3.2 has been revised. It now reads: *"The estimated fraction of evaporation FE at each soil depth was derived by means of an evaporative enrichment of heavier isotopes in the soil water. Fig. 5 shows FE estimated based on both isotopes for the growing seasons WS and DS, the growing stages GS1-GS3, the different crops wet rice, dry rice, and maize as well as for different soil depths. A clear trend of FE with soil depth can be depicted at all growing stages during the DS for both crops, maize, and dry rice, reaching below 0.2 for dry rice and even below 0.1 for maize in deep soils (Fig. 5g-l). During the DS, soils in dry rice fields showed high soil FE at shallow depths at the beginning of the first two growing stages GS1 (0.54±0.1) and GS2 (0.50±0.1) which decreased to FE of around 0.27±0.1 at GS3. Further, we observed lower FE average values (0.2±0.1) in deep soils between 0.25 and 0.6 m in these fields. For maize, FE remained stable at 0.3±0.1 in shallow soils throughout the season and decreased with depth for both isotopes (to around 0.07±0.05) (Fig. 5j, k, l). The FE in shallow soils of wet rice in the DS ranged from 0.42±0.08 to 0.20±0.08 (similar for both isotopes) and remained nearly stable in deep soils at 0.13±0.1 (Fig. 5d, e, f). Overall, we did not find a similar decreasing trend with depth, as reported for dry rice and maize. Instead, particularly during GS2 and GS3, the highest fractions of FE were found at moderate soil depths of 0.1 to 0.2 m. Results regarding the estimation of FE based on δ2H and δ18O are fairly similar for all dry season crops. For wet rice in the WS, FE of both isotopes differed significantly by 0.1 to 0.2 for the top soil and even 0.5 for deep soils (Fig. 5a, b, c). During the WS, FE in shallow soil decreased from around 0.72±0.1 (GS1) to 0.47±0.06 (GS3) for δ2H and from 0.87±0.07 (GS1) to 0.76±0.07 (GS3) for δ18O. The general trend with slightly higher FE at moderate soil depths and again decreasing FE further down, which we observed for wet rice during the dry season, was also confirmed for wet rice during the wet season. The soil water in wet rice fields during the WS carried a larger signal of high evaporation losses down along the soil profile. The estimated FE from ponding surface water (data not shown in Fig. 5)*

*was found to be larger during the WS than during the DS with no significant difference between δ2H and δ18O. The FE of ponded water during the WS did not fluctuate with time, and remained close to 0.92±0.07, while during the DS values decreased from GS1 (0.67±0.03) to GS3 (0.24±0.01). Here, FE of ponded surface water indicates a high evaporation loss during the WS. The evaporation signal is carried to deeper layers by subsequent infiltration and percolation. "*

P8L9-13: A figure of F_E for ponded water would help illustrating this description and the related discussion.

Surface ponded water is valid only for wet rice and therefore, we could only plot 6 values in total. Therefore, we do not think it is necessary to include such a figure.

P8L13: It would be more correct to say "by subsequent infiltration and percolation.".

Added as recommended.

P8L16: Only if lateral transfers (and return flow) can be neglected, and if root uptake is assumed as being non-fractionating. I suggest rephrasing as follows: "In the absence of lateral water transfers and assuming negligible fractionation from root water uptake, the isotopic profiles in soil water reflect a balance between mixing from infiltration and percolation, and fractionation from soil evaporation."

Rephased as recommended.

P8L28-30: Shouldn't this sentence should open the next paragraph, as it related to flooded conditions? Also, to be clearer I suggest moving "in flooded fields" to the beginning of the sentence.

During revision of the manuscript, we deleted this sentence.

P9L3-10: This is certainly an interesting interpretation. As I understand it, the shallowest soil samples (is it 0m?) are very depleted, meaning there are already beyond the infiltration front, above which the "single water column" including ponding water is affected by fractionation? How can the infiltration be so (infinitely) shallow?

The most enriched point (which we refer here as infiltration front) is at 5 cm or below than that for wet rice (which goes until 20 cm at some cases (especially at GS3). Therefore, it is not infinitely shallow.

Further, we deleted the sentence regarding the single compartment, as it was misleading.

P9L19-23: The authors start by mentioning hydraulic redistribution a potential factor to isotopic profiles, cite review works and then finally state it is not important, based on reference that few can check (I, for example, cannot read German)...

It is not really convincing, especially given the fact that the authors do not give rough estimates for transpiration fraction or root profile.

I suggest finding a better explanation and basis in the literature, or else acknowledge that the potential impact of hydraulic redistribution is unknown and should be the focus of further studies.

We added the following sentence at the end of the mentioned section to address the point. "*While the review of (Walter, 2010) only indicated a limited impact of hydraulic redistribution on the isotopic composition of soil water, the selective removal of water combined with redistribution can be relevant. Still, isotopic measurements alone are not sufficient to estimate redistribution volumes (Emerman and Dawson, 1996) and therefore the potential impact of hydraulic redistribution requires a combination of physical transport modelling and isotopic composition and should be the focus of further studies.*"

P10 L5-10: This description should in the Result section, not in the Discussion.

We moved few sentences from the mentioned section to the results and kept some with some edits which we want to focus in the discussion.

P11L2-3: By "plant water and rainwater", do the authors mean "the isotopic composition of xylem water and rainfall"? Please clarify.

Yes, we made it clearer.

P11L24: This is not supported by a growing body of litterature showing that plant transpiration can be a fractionating process (e.g. Vargas et al., 2017; Barbeta et al., 2019, Poca et al., 2019).

We removed that part to avoid the complication of this topic.

P12L14: "suggesting" would be more accurate than "stipulating".

Edited as recommended.

Fig3: Having the lc-excess values for ponding and irrigation water in subplots d and h would help the discussion.

Added as recommended.

Technical comments

P1L15: Water has no isotopes, only isotopologues. Please consider using "stable isotopes in water"?

The most used term in the literature is "stable isotopes of water" and as we are referring to the isotopes of water, we agree with it (google hits for higher "stable isotopes of water" = 313,000).

P1L16: typo: Craig-Gordon model

Corrected. Thank you.

P1L28: "ideal" is subjective, consider removing it.

Removed as suggested.

P6L16-17: typo: \delta_P is the signal of precipitation water.

Can't find it.

P6L16-17: "Between" is more grammatically correct than "in both"

Corrected as suggested.

P6L22: "Soil water from crops" : do the authors mean "soil water below crops"?

Yes, corrected.

P6L27: Maybe "decreased again until about 0.2 m" instead?

Changed as suggested.

P7L7-8: I suggest "as plants were growing, while such clear patterns were not be observed"

Changed as suggested.

P8L26: grammar: "this leads to the accumulation of..."

Corrected as suggested.

P8L28-29: missing word and rewording: "in soils beneath dry rice and maize"

Added.

P9L2-3: It would more correct to say that "water is enriched in heavier isotopes as depth increases" or "the concentration in heavier isotopes increases with depth"

Edited as recommended.

P10L5: "increasing negativity" sounds odd, I suggest "increasingly negative values"

Changed as suggested.

P10L5-6: I suggest "across growth stages" instead of "along the growth"

Changed as suggested.

P10L20: typo: the correct reference is "Allison (1982)"

Corrected.

P12L30-32: It would be more correct to say "We also quantified the relative fraction of soil water returning to the atmosphere as direct evaporation, and related its pattern to crop types and seasons".

Changed as suggested.

P12L32: English: "would be needed" instead of "would be highly appreciated"

Edited as suggested.

Fig3: Just like most readers (I think), I would appreciate a higher-definition figure.

High resolution figures were submitted as PDF and those will be added to the final version.

Also, the choice of colour for RW makes it hard to distinguish from GS2 (and GS3).

Colour of the RW was changed. However, the colour difference is much clear in the high-resolution figure.

Fig4: I suggest plotting the regression lines behind the individual soil values, and for example in black, to better see the depth-coloured soil values. Also, why not plotting the individual isotopic value for rain and irrigation water?

We changed the plot as recommended. However, plotting individual values of IW and RW makes the plot bit messy as it overlaps with soil values. Therefore, we prefer to keep as it is.

[revised manuscript text omitted]

---

## Author Response (AR4)

Dear editor,

We would like to thank you for the valuable feedback provided for our manuscript entitled, "Investigating unproductive water losses from irrigated agricultural crops in the humid tropics through analyses of stable isotopes of water".
Please find our point-by-point responses (in blue) to the comments (in black) below.

We believe that the modifications result in an improved manuscript. We hope that the manuscript is now suitable for consideration for publication as a research paper in Hydrology and Earth System Sciences.

We look forward to hearing from you.

Best regards,

On behalf of the authors,

Amani Mahindawansha

######################################

Dear first author, Amani Mahindawansha, and corresponding author Lutz Breuer,

Thanks for sending the revised manuscript and I appreciate the responses provided to each of the comments.I think that the manuscript can be published after clarification of the two following issues that came up.

1.) Now that I see can in Figure 3 that RW isotope ratios are more enriched than the soil water isotopes for Wet rice (WS) and GS2 and GS3 (Fig. 3 a&b) I do not understand how one gets the high FE values reported in Figure 5. In this case, dP will be bigger than dS in Equation (1) and that would result in a negative FE value. In general, it would not be expected that the fraction of evaporation loss is highest for the soil water samples that are least fractionated (Wet rice (WS)). Please look into your calculations and explain what is going on here – I seem to misunderstand something.

Thank you for the comment. As shown in Figure 4, only for Wet rice (WS) at GS2 and GS3 the isotopic composition of soil water is more depleted than the weighted isotopic composition of rainwater and of irrigation water. For Wet rice (WS) at GS2 and GS3 the weighted isotopic composition of precipitation ($\delta^{18}O$ -4.42 ‰ and $\delta^2H$ -26.82) does not reflect the true input end member. The isotopic composition of infiltrated water has a negative bias compared to the weighted isotopic composition of rainwater. Therefore, as mentioned in the text, "During the WS, $\delta p$ was estimated as the weighted average of the isotopic signals from the frequently occurring large precipitation events (larger than 10 mm)." The values used in equation 1 are -36.6 ‰ for $\delta^2H$ and -5.8 ‰ for $\delta^{18}O$. As a result, $\delta p$ is lower than $\delta s$ and the FE value is positive.

There is also an effect of the isotopic composition of atmospheric vapour ($\delta_A$) and of relative humidity on the fractionation. The isotopic composition of atmospheric vapour ($\delta_A$) was calculated assuming equilibrium with precipitation (Eq. 4 in the supplementary materials). A more depleted atmospheric moisture during the wet season increases the estimated value of FE.

Thank you for bringing this point up. We agree that it is better to clarify the calculation method and resulting values in the text. Please see P5L24-25, P8L6-9, and P8L28-33 in the revised manuscript.

2.) Figure 2: I understand that in hydrological manuscripts, precipitation on bar plots often has an inverse axis. However, this is usually only necessary if there is also stream discharge or similar data additionally plotted in the graph. More importantly, the y-Axis for d2H is inverse, which is very confusing and should be changed. I am sorry that I did not realize in the previous round of reviews.

We agree. We changed the figure as recommended.